# A comprehensive map of alternative polyadenylation in African American and European American lung cancer patients

Adriana Zingone [1], Sanju Sinha[1,2], Michael Ante [3,4], Cu Nguyen[5], Dalia Daujotyte[3], Elise D. Bowman[1],
Neelam Sinha [2], Khadijah A. Mitchell [1], Qingrong Chen[5], Chunhua Yan[5], Phillipe Loher[6],
Daoud Meerzaman[5], Eytan Ruppin[2] & Bríd M. Ryan [1]✉

Deciphering the post-transcriptional mechanisms (PTM) regulating gene expression is critical to understand the dynamics underlying transcriptomic regulation in cancer. Alternative polyadenylation (APA)—regulation of mRNA 3′UTR length by alternating poly(A) site usage —is a key PTM mechanism whose comprehensive analysis in cancer remains an important open challenge. Here we use a method and analysis pipeline that sequences 3′end-enriched RNA directly to overcome the saturation limitation of traditional 5′–3′ based sequencing. We comprehensively map the APA landscape in lung cancer in a cohort of 98 tumor/non-involved tissues derived from European American and African American patients. We identify a global shortening of 3′UTR transcripts in lung cancer, with notable functional implications on the expression of both coding and noncoding genes. We find that APA of non-coding RNA transcripts (long non-coding RNAs and microRNAs) is a recurrent event in lung cancer and discover that the selection of alternative polyA sites is a form of non-coding RNA expression control. Our results indicate that mRNA transcripts from EAs are two times more likely than AAs to undergo APA in lung cancer. Taken together, our findings comprehensively map and identify the important functional role of alternative polyadenylation in determining transcriptomic heterogeneity in lung cancer.

[1] Laboratory of Human Carcinogenesis, Center for Cancer Research, National Cancer Institute, Bethesda, MD 20892, US. [2] Cancer Data Science Laboratory, Center for Cancer Research, National Cancer Institute, Bethesda, MD, US. [3] Lexogen GmbH, Campus Vienna Biocenter 5, 1030 Vienna, Austria. [4] Ares Genetics GmbH, Karl-Farkas-Gasse 18, 1030 Vienna, Austria. [5] Computational Genomics Research, Center for Biomedical Informatics and Information Technology (CBIIT), National Cancer Institute, 9609 Medical Center Drive, Rockville, MD 20850, US. [6] Computational Medicine Center, Sidney Kimmel Medical College, Thomas Jefferson University, Philadelphia, PA 19017, US. ✉email: brid.ryan@nih.gov

Precision medicine in cancer management often relies on the use of biomarkers to classify molecular subtypes and stratify responder and non-responder patients. However, transcriptomic-based signatures can be confounded by molecular intratumor heterogeneity, as found in multiple cancer types[1–3], including lung cancer[4], which is the leading cause of cancer-related death in the United States[5]. Post-transcriptional modification (PTM) is a major contributor to transcriptomic heterogeneity. Global regulation of PTM is evident in many immune cells, including T-cells, where it is involved in cell proliferation, differentiation, and response to extracellular stress[6]. Tumor cells can also rewire the transcriptome via PTM making it important to understand the dynamics underlying transcriptomic regulation in cancer cells.

Almost all eukaryotic mRNAs undergo polyadenylation, a nuclear process that involves the addition of non-templated adenosines to the 3′ end of transcripts. Approximately 70% of human genes contain more than one polyadenylation site (polyA site, PAS). The use of different or alternate PAS is called alternative polyadenylation (APA). APA is a key PTM mechanism regulating nuclear export, stability, and translational efficiency of mature mRNAs[7–10] and leads to multiple forms of the transcript with different 3′ untranslated region (UTR) lengths[10–12]. This A/U rich 3′ UTR is a prominent docking site for RNA regulatory elements, including miRNAs and RNA binding proteins such as Hu-Antigen R (HuR), also known as ELAV like RNA binding protein-1 (ELAV1), through which it regulates transcript stability, transcript export, and cellular localization and protein translation[13–15]. Further, DNA-encoded SNPs located in the mRNA 3′UTR may not be functional anymore if 3′UTR length is shortened[16].

The selection of a polyA site is a dynamic process that contributes to both normal physiology and pathological phenotypes[10,11,17–20]. For instance, during B-cell differentiation, the use of an intronic PAS generates a secreted form of the IgM protein while the use of a distal site on the IgM transcript generates a membrane-bound form[17]. Furthermore, an inherited mutation that drives the use of a proximal polyA site in the TNSFRS2 gene—which encodes the cytokine BAFF—leads to increased BAFF expression and susceptibility to autoimmune diseases[20]. Dysregulated APA is also linked with cancer. Sandberg and Mayr independently demonstrated a relationship between APA with proliferation and with carcinogenesis[11,21], with recent studies highlighting global shortening of 3′UTR as a characteristic of the cancer transcriptome[11,21].

Although a few studies have assessed APA in lung cancer, they relied on conventional methods largely based on 5′–3′ RNA-seq data. This approach can lead to saturation[22–24] as the proportion of final reads from a polyA transcript/fragment using either polyA selected RNA-seq or total RNA-seq can be very low[25–27]. This underscores the need to comprehensively and specifically map polyA sites and APA in cancer with a method that overcomes this saturation. Since we previously observed that lung tumors derived from European Americans (EAs) are enriched in cell proliferation and cell cycle pathways compared with African American (AAs)[28], we designed our study to investigate whether differences in APA between the two populations exist. Here, using 3′end-enriched RNA and direct mapping of polyA sites, in contrast to traditional 5′–3′ sequencing methods, we identify dynamic APA events and increase both the depth and accuracy of the analysis. We find that lung cancer cells are significantly more likely to produce mRNA transcripts with shorter 3′UTRs than normal cells and that these shorter transcripts are related to poor patient survival. In this study we show that global shortening of 3′ UTR transcripts is present in lung cancer, selection of alternative polyA sites in both long non-coding and microRNA is a form of non-coding RNA expression regulation, and mRNA transcripts from EA are two-fold more likely than AA to undergo APA in lung cancer.

## Results

### Identification, quantification, and validation of polyA sites (PAS).
The 3′UTR-seq method generated a nucleotide-level resolution map of how cells use distinct polyA sites in mRNA transcripts in lung cancer. We identified 123,475 PAS, of which 102,005 were annotated to known genes (Ensembl v90). To validate detected PAS, we compared our data with known polyA databases where our data mapped to 79% and 86% of the Derti[29] and Gruber[30] databases, respectively (Fig. 1a), indicating that our analysis compared well with previously published data and thus ensuring the quality. We next used External RNA Controls Consortium (ERCC) spike-in RNAs to evaluate diagnostic performance, limit of detection of ratio estimates, and expression ratio variability. All aspects of our experiment's technical performance passed these quality control measures (Supplementary Fig. 1). Following several detailed filtering steps (see "Methods" section), 37,037 polyA sites annotated to 17,220 genes remained. Of those 17,220 genes, 7870 (46%) had more than one polyA site (Supplementary Data 1) (Fig. 1b), which is lower than the 70% reported by other studies and is likely due to the stringent filtering approach we applied. Although the majority (79%) of distal sites were within the 3′UTR as expected, only half of the proximal polyA sites were, with the rest in introns (35%) or exons (12%) (Fig. 1c).

### Characterization and classification of polyA sites in lung cancer.
To examine the functional status of polyA sites in lung cancer, we focused on the 7,870 genes for which we identified multiple polyA sites. Of these, 3,531 genes (44%) had statistically significant changes (switches) in polyA site usage between tumor and adjacent non-involved tissue; 3,119 had enhanced proximal site usage and 412 had enhanced distal site usage in tumor relative to adjacent non-involved tissue (Fig. 2a, b and Supplementary Data 2–4). Thus, consistent with other tumor types[21,31], mRNAs are significantly more likely to produce transcripts with shorter 3′UTRs compared with non-transformed cells.

Of the 3,119 transcripts with enhanced use of proximal PAS in tumor compared with adjacent non-involved tissues, 1,549 were same-exon, 903 were composite-exon and 667 were skipped-exon pair types. In contrast, among the 412 genes with significantly increased usage of distal sites, 81 were same-exon, while 233 and 98 were composite-exon and skipped-exon, respectively (Supplementary Data 2 and 3). These results are consistent with previous studies indicating that a switch in the usage of proximal polyA sites in cancer involves predominantly same-exon sites[31]. Many proximal polyA sites with increased use in tumor tissues were located in intronic ($n = 925$) and exonic ($n = 301$) regions (Supplementary Data 4) suggesting that use of these sites could lead to truncated proteins with different amino acid sequences and/or function. There was no enrichment in sense or anti-sense transcripts among regulated APA sites (Fig. 2c).

For each gene, we also defined a measure of shortening, the polyA site index (PSI), which is a ratio of read number from a proximal site divided by the total number of reads from proximal and distal sites. The ΔPSI captures the relative use of proximal sites compared with distal sites in tumor cells compared to non-involved tissues and confirms that 3′UTR lengths are globally shortened in lung cancer cells (Supplementary Data 5) (Supplementary Fig. 2a). Using genome-wide RNA-seq reads from the Quantseq output (see "Methods" section), we observed that expression of key genes involved in the regulation of polyadenylation, including CPSF and

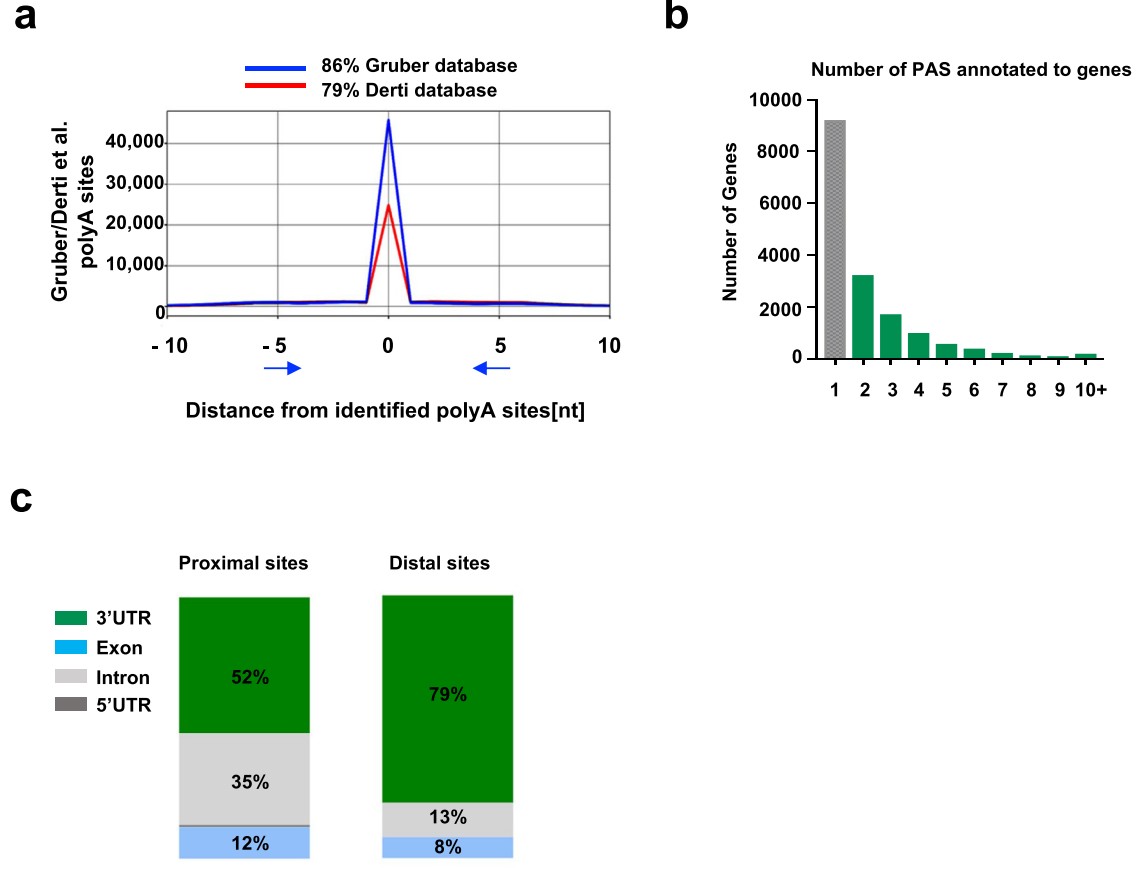

**Fig. 1 Identification, quantification, and validation of poly (A) sites from QuantSeq. a** Overlap of polyA sites (*y* axis) with previously published databases (Gruber et al.[69] [https://www.ncbi.nlm.nih.gov/sra/docs/] SRP065825 and Derti et al. [29] [https://www.ncbi.nlm.nih.gov/geo/query/acc.cgi?acc=GSE30198]) assuring a high quality. Percentages in the legend are reported with spacing −5 to +5 nucleotides around identified PAS. **b** Detected polyadenylation sites across genes: 37,037 poly (A) sites annotated to 17,220 genes, 46% of genes possess more than 1 site (green bars). **c** Distribution of proximal and distal site location within the gene.

*CSTF*, were upregulated in tumors compared with non-involved lung tissues suggesting a possible mechanism of 3′UTR mRNA shortening (Fig. 2d).

Although we observed some differences in the specific transcripts undergoing 3′UTR shortening in lung adenocarcinoma (LUAD) and lung squamous cell carcinoma (LUSC) (Supplementary Data 6) (Supplementary Fig. 2b), overall, 3′UTR length was similar between LUAD and LUSC (Supplementary Fig. 2c).

**Molecular and clinical features of lung cancer associated with APA.** To characterize the pathways that are targeted by APA in lung cancer, we performed a pathway enrichment analysis using IPA[32] and found those mRNA transcripts with shorter 3′UTRs were enriched with cell cycle/proliferation pathways often dysregulated in cancer, including mTOR and its target ubiquitin proteolysis pathway, receptor tyrosine kinase signaling and signaling specific to lung cancer. The mTOR finding is consistent with recent evidence identifying activation of the mTOR pathway as a driver of APA in cancer[33]. Transcripts with longer 3′UTRs were generally enriched in metabolism and p53 signaling-related pathways (Fig. 2e and Supplementary Data 7) collectively suggesting that APA contributes to the molecular features of lung cancer.

Stress and exposure to various environmental factors can modulate APA[34,35], therefore we asked whether exposure to tobacco could modulate APA in lung cancer. We compared genes

with regulated polyA sites in tumor tissue only between current and former smokers but did not detect any significant changes (Supplementary Fig. 3) suggesting that smoking is not a major driver of APA events in lung cancer.

As cancer cells favor the selection of shorter 3′UTRs, we reasoned that the trimming of mRNA transcripts could be a prognostic biomarker. To determine whether APA captures genes with clinical relevance, we performed a multivariate Cox regression correcting for patient age, sex, race, and tumor stage for each gene and identified a strong prognostic signature based on the APA usage pattern of 12 genes in lung cancer (Fig. 2f and Supplementary Data 8), including the type I transmembrane glycoprotein immunoglobulin *CD96* and a key homologous recombination repair gene *POLA2*. Further, gene ontology analysis of APA events associated with survival revealed enrichment of cellular metabolic processes (Supplementary Data 9).

**Genes with 3′ shortening have a higher concordance between mRNA and protein levels.** When a miRNA engages with its cognate miRNA binding site, it can induce message degradation or destabilization. Thus, escape from miRNA repression can result in a more stable transcript with a longer half-life and relative higher abundance of the mRNA and resulting in more protein[11,21]. Alternatively, mRNA levels are not always affected given that most miRNA/mRNA binding interactions are imperfect and do not lead to mRNA degradation. Lastly, the competing-endogenous (ceRNA)

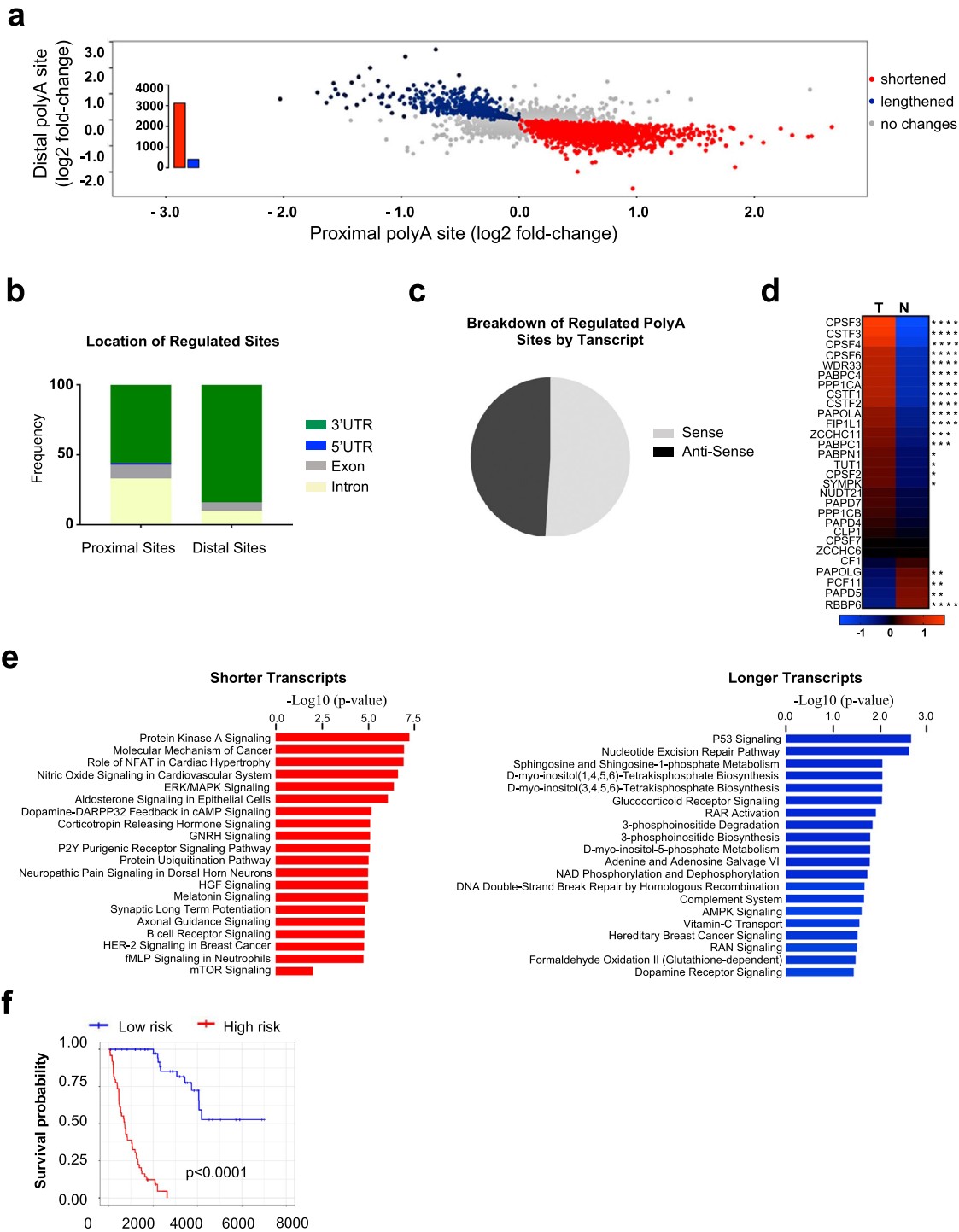

**Fig. 2 Role of alternative polyadenylation in shaping the molecular and clinical features of lung cancer. a** Global shortening of 3′UTR detected by DexSeq2 analysis. The dot plot map shows genes undergoing significant 3′UTR shortening (red) or significant 3′UTR lengthening (blue) and no changes (gray) in tumors compared with adjacent non-involved tissues. Each dot corresponds to a gene. **b** Genomic location of regulated proximal and distal sites in tumors. **c** Breakdown of regulated sites by sense/anti-sense strands. **d** Expression of polyadenylation-related genes in tumor and adjacent non-involved tissues. Heatmap of combined z-score and gene expression analysis of 29 known polyadenylation factors in tumors compared to non-involved tissues. The asterisk indicates the degree of significance of differentially expressed genes. **e** Enriched pathways. Pathway significantly enriched have a −log10 (p-value) greater than 1.3 (p-value < 0.05; Fisher's exact test; one-tailed test). Ingenuity canonical pathways analysis in genes with enhanced proximal and distal sites. **f** Relationship between 3′UTR shortening and lung cancer survival. A two-sided log-rank test is used to compare the survival times between two groups. The p-value of the log-rank test statistic is commonly approximated by the chi-square distribution and thus only approximate in a higher limit. *p < 0.05; **p < 0.01; ***p < 0.001; ****p < 0.0001.

hypothesis argues that, rather than affecting mRNA expression *in cis*, APA impacts mRNA *in trans* by altering the stochastic nature by which the loss of one miRNA binding site frees up miRNAs to bind to other targets[14,19–21,36,37]. To test these hypotheses in lung cancer, we first found that among genes with significant 3′UTR shortening, there was an average loss of nine miRNA binding sites per gene (range 1–51), with enrichment for lung cancer-associated miR-124, miR-181, let-7, and miR-27 binding sites (Supplementary Fig. 4a, b and Supplementary Data 10, 11). We calculated a gene-wise correlation between PSI and expression and found that the PSI of 1,376 of the 3,531 genes significantly correlates with expression (Supplementary Data 12), roughly evenly split between positive and negative associations. We then asked, what is the probability of observing this number of genes ($N = 1,376$) or greater by chance. To test this, we shuffled the APA matrix 10,000 times and calculated an empirical significance ($p < 0.17$) by counting the number of times the genes with Spearman Rho > 0.1 are greater than or equal to 1,376.

To test whether APA could drive a higher concordance between mRNA-protein levels for genes with shorter 3′UTRs, we mined cancer cell line encyclopedia (CCLE) proteomics and RNA-seq data[38]. Using RNA-seq reads, we inferred the PSI as before[24] and found that genes with higher median PSI (more shortening) have a stronger correlation between their mRNA and protein (top 10% vs bottom 10% genes ranked by median PSI, Wilcoxon rank-sum $p = 0.027$) (Supplementary Data 13 and Supplementary Fig. 5). We replicated this observation at a tumor tissue level using TCGA[39] and CPTAC data[40,41] where mRNA levels and protein abundance for the same samples are available [see "Methods" section] (Wilcoxon rank-sum $p = 0.03$ [breast] and $p = 0.075$ [lung]) (Supplementary Data 13 and Supplementary Fig. 5). These data suggest that the tighter correlation between mRNA and protein found in both cell lines and tumor tissues is, at least partially, due to 3′UTR mRNA shortening in cancer cells.

**Identification of recurrent cancer-related APA events in non-coding RNAs.** Given the evidence that non-coding RNAs also undergo polyadenylation, we proposed, and subsequently identified, significant APA events in non-coding RNAs. Overall, 955 (12%) of the transcripts with more than one PAS in our analysis mapped to a non-coding RNA (Supplementary Data 4). Of the 27 miRNA host genes detected by our sequencing method and that passed the read count threshold, 14 had more than one PAS and, of these, 10 (71%) underwent significant alternative polyadenylation in lung cancer, including miR-155 and let-7B (Supplementary Data 4). Furthermore, our method detected 928 long non-coding RNAs, of which 231 had more than one polyA site. Of these, 57 (25%) underwent alternative polyadenylation in lung cancer, including DLEU1, LINC01138, and PVT1 (Supplementary Data 4).

To test whether APA of non-coding transcripts modulates expression, we generated corresponding total RNA-seq data and small RNA-seq data for 39 tumor/non-involved tissue pairs where we had sufficient tissue and compared the average difference in tumor and adjacent non-involved PSI to the average difference in tumor and adjacent non-involved miRNA expression. Of the 10 miRNA host genes that underwent recurrent APA in lung cancer, two (miR-3936, miR-646) did not have mapped reads in the small RNA-seq file. Of the remaining eight, host miRNA APA was correlated with the expression of the individual miRNAs residing within the host gene (Fig. 3a and Supplementary Data 14). Interestingly, cancer-related APA can have a bi-directional effect on mature miRNA expression. For example, decreased use of a proximal polyA site on the miR17HG

transcript, which includes miR-17, miR-18a, miR-19a, miR-19b, and miR-92a, is associated with increased mature miRNA expression in tumors compared with adjacent non-involved tissues (Fig. 3a). However, in LET7BHG and miR-155, increased use of a distal polyA site is associated with decreased mature let-7b expression in tumors compared with adjacent non-involved tissue. We also analyzed the impact of APA on isomiR expression and observed similar trends (Supplementary Data 14). Several of the long non-coding RNAs that underwent APA in lung cancer, including DLEU1 and PVT1 which are associated with lung cancer survival[42], also correlated with expression (Fig. 3b and Supplementary Data 15). We also tested the performance of the prognostic index separately in EAs and AAs and computed the hazard ratio (HR) for each population separately (AA HR = 4.1, $p < 0.001$; EA HR = 4.3, $p < 0.001$).

Interestingly, the regions of these non-coding transcripts that were retained or lost due to APA in cancer cells frequently overlapped with HuR/ELAV1 binding regions. Further, by mining RIP-seq data we observed significant differential binding between HuR to the regions retained or lost through APA in both long non-coding RNAs and microRNAs (Fig. 3c). As HuR modulates transcripts—through both stabilization[43] and destabilization[44], gain or loss of RNA bindings proteins is a potential mechanism by which APA of non-coding RNAs controls expression of mature transcripts. For example, increased use of a distal polyA site on the miR17HG RNA transcript relative to a proximal one (Supplementary Data 4) leads to the inclusion of a region with multiple binding sites for HuR proteins in tumor tissues (Fig. 3a). Indeed, our data show that there is more HuR binding to this region in cancer, compared with normal cell lines (Fig. 3c). As the PSI correlates with increased miR-17 cluster expression, we speculate that this HuR binding leads to greater transcript stabilization, greater processing, and, therefore, higher expression.

Collectively, these data suggest that APA of non-coding RNA transcripts is a recurrent event in lung cancer and that the selection of alternative polyA sites could be a form of non-coding RNA expression control.

**Population differences 3′UTR length.** In a previous transcriptomic analysis of lung cancer in European Americans and African Americans, we observed that genes dysregulated in tumors from EAs only are enriched in cell proliferation and cell cycle pathways compared with AAs[28]. Given that proliferating cells are more likely to have shorter 3′UTRs[11], we tested the hypothesis that lung tumors from EAs would be enriched in regulated gene-level APA events and have global shorter 3′UTRs compared with AAs. When compared with non-involved adjacent tissues, tumor cells from EAs had more significant APA events compared with AAs (Fig. 4a). Specifically, of genes with more than one APA site, 30% of genes (2,276) undergo 3′UTR shortening in EAs compared with 17% (1,360) in AAs, following Benjamini-Hochberg FDR-adjusted *p*-values for each site. This represents a greater than 2-fold increase of APA events in EAs (Supplementary Data 16 and 17). We repeated this analysis, comparing the gene-wise number of APA events (log2PSI value) in tumor vs non-involved adjacent tissues in both EAs and AAs using a multivariate regression correcting for patient age, sex, tumor stage, and smoking status and consistently found a 2-fold higher number of genes going through APA events in EAs vs AAs. There was no specific enrichment in LUSC vs. LUAD (Supplementary Fig 2d).

As no other large cancer datasets with 3′UTR-specific sequencing exist to our knowledge, we mined TCGA 5′–3′ RNA-seq reads from non-small cell lung cancer[24] to test whether EAs have more global

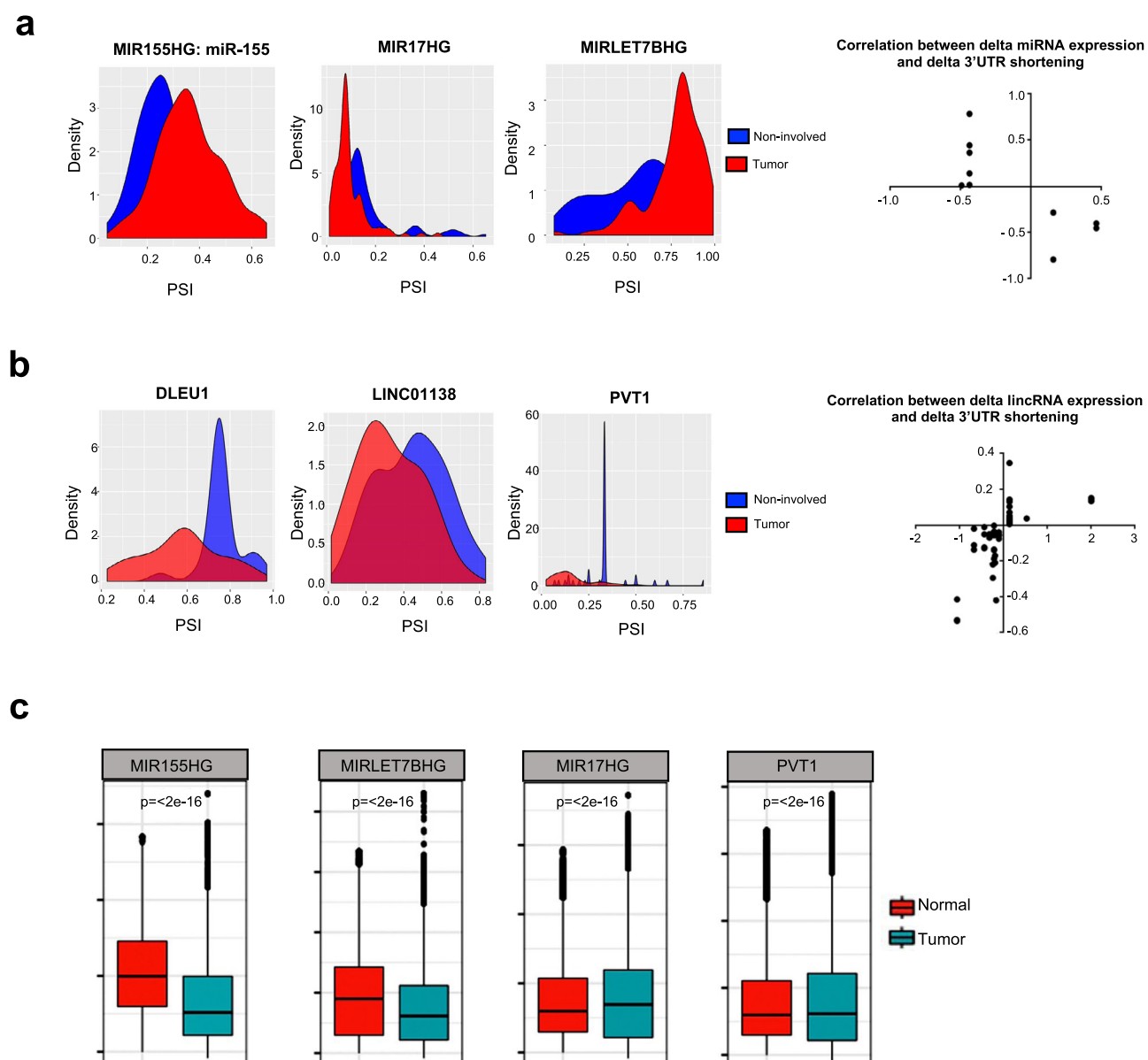

**Fig. 3 Alternative polyadenylation of non-coding RNAs in lung cancer and relationship with expression in microRNAs long non-coding RNAs. a** Panels show the correlation between the average 3′UTR length difference in tumor and adjacent non-involved tissues (PSI) to the average miRNA expression difference in tumor and adjacent non-involved tissues. In the scatter plot, the *y* axis denotes the delta PSI and the *x* axis denotes delta miRNA. **b** Panels show the correlation between the average 3′UTR length difference to the average long non-coding RNA expression difference in tumor and adjacent non-involved tissues. In the scatter plot, the *y* axis denotes the delta PSI and the *x* axis denotes delta long-non-coding RNA. **c** Difference in HuR/ELAV1 binding to mRNA segments gained and lost through APA based on RNA immunoprecipitation (RIP-seq) assays profile measuring ELAV1/HuR RNA binding of K562 leukemia cells and normal GM12878 cells as a control from GSE35585. For a given gene or element, ELAV1/HuR binding is quantified by counting the number of reads between primary proximal and distal polyA site of usage in both cell lines individually. Here in the box plot, the lower and upper hinges of the boxes correspond to the 25th and 75th percentiles and the whiskers represent the 1.5× inter-quartile range (IQR) extending from the hinges. The center lines denote the median and the black line represents the rest of the distribution, except for the points that are determined to be "outliers". A two-tailed Wilcoxon rank-sum test is used to compute the significance of the difference between medians. The *p*-values are below the minimum floating integer that can be computed using standard *R* and thus the exact value reported by *R* is 0. Thus, we provide the approximate upper bound reported by *R*.

3′UTR shortening compared with AAs. We calculated a PSI[24] for each gene and using a similar multivariate regression above, tested the association between PSI and race correcting for sex, tumor stage, and age of the patient (this analysis was done in tumors only as there were not sufficient adjacent non-involved samples from AA patients for comparison to EAs). We confirmed that while more genes undergo shortening in EA tumors in comparison to AA tumors from LUSC (EA = 1,254 compared with AA = 598) and LUAD (EA = 1,443 compared with AA 1,075) (Supplementary

Fig. 6b), the global difference in 3′UTR length, considering tumor tissue only, is not significantly different (Fig. 4c). To test pan-cancer population differences in APA, we next mined the same data set for 11,265 TCGA samples from 28 cancer types for 7,062 genes[45]. Again, we modeled the PSI index as a function of race correcting for cancer type, sex, tumor stage, and age of the patient. Consistently, we found a higher number of genes with 3′UTR shortening in EA tumors compared with AA tumors in pan-cancer (in EA = 1,643 compared to in AA = 1,277) (Fig. 4b). Some of the strongest

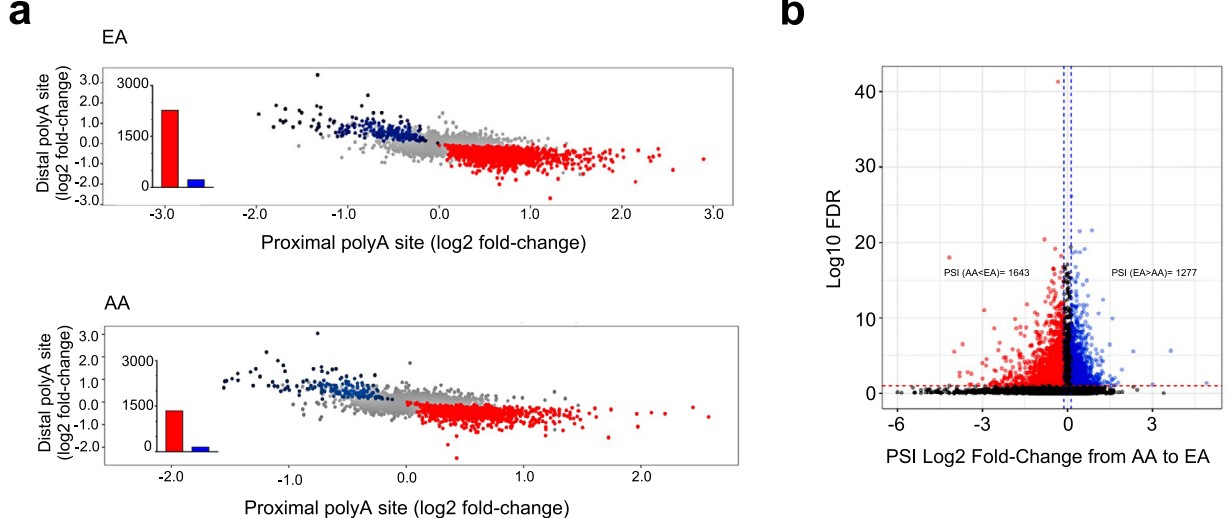

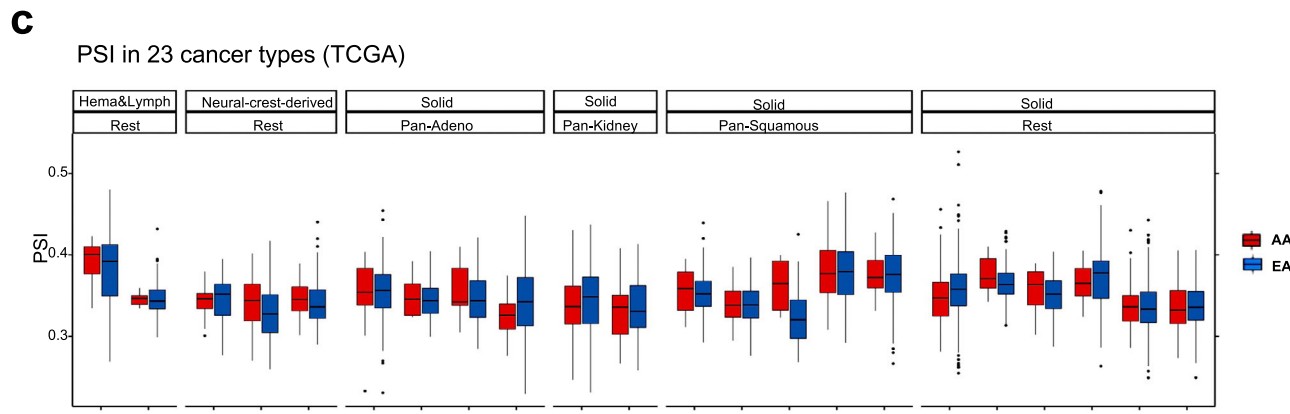

**Fig. 4 Race-related differences in global 3′UTR length of genes across multiple cancer types. a** Distribution of shortening (red) and lengthening (blue) events in lung cancer in EAs and AAs. **b** Pan-cancer population differences shortening (red) and lengthening (blue) events in EAs and AAs. **c** Comparison of PSI in EAs and AAs in tumor tissues in TCGA. First, cancer types are categorized by cell type or tissue of origin, if possible, where defined groups are pan-squamous (squamous cell-derived tumors), pan-adeno (glandular structures in epithelial tissue derived tumors), pan-kidney (tumors originating in the kidney), and rest (referring to cancer types that cannot be categorized and include LAML, THYM, GBM, LGG, SARC, BRCA, LIHC, OV, TCGT, THCA, and UCEC; refer here for reference to each cancer type: https://gdc.cancer.gov/resources-tcga-users/tcga-code-tables/tcga-study-abbreviations). Second, additional categorization was performed based on tissue type (where solid is derived from solid tumors and neural-crest and Hema & Lymph—hematologic and lymphatic tumors). A two-sided Wilcoxon Rank-sum test has been performed within each cancer type and significance before multiple testing correction is provided. LAML $p = 0.55$; THYM $p = 0.92$; GBM $p = 0.55$; SARC $p = 0.15$; LUAD $p = 0.64$; PAAD $p = 0.79$; PRAD $p = 0.42$; STAD $p = 0.11$; KIRC $p = 0.069$; KIRP $p = 0.756$; BLCA $p = 0.332$; CESC $p = 0.816$; ESCA $p = 0.031$; HSNC $p = 0.977$; LUSC $p = 0.703$; LUSC $p = 0.703$; BRCA $p = 0.00011$; LIHC $p = 0.14103$; OV $p = 0.14827$; TGCT $p = 0.65657$; THCA $p = 0.70424$; UCEC $p = 0.73852$. Here in the box plot, the lower and upper hinges of the boxes correspond to the 25th and 75th percentiles and the whiskers represent the 1.5× inter-quartile range (IQR) extending from the hinges. The center lines denote the median and the black line represents the rest of the distribution, except for the points that are determined to be "outliers". A two-sided Wilcoxon rank-sum test is used to compute the significance of the difference between medians.

differences were observed in breast ($p = 0.0001$) and esophageal cancer ($p = 0.031$), where global shortening of mRNA transcripts was also observed (Fig. 4c) (Supplementary Data 18).

As demonstrated earlier (Fig. 2d), there is global dysregulation of proteins involved with APA in cancer, i.e., the PA machinery. To determine whether the population differences in APA are related to differential activity of PA machinery between EA and AA tumors, we computed an PA machinery activity score (median scaled expression of PA machinery genes and found it to be positively correlated with PSI-load (Rho = 0.18, $p < 2E−16$)), suggesting that expression of machinery-related genes modulates

global APA changes. We then computed the difference in the PA machinery activity score by race in each cancer type in TCGA and observed a significantly higher activity in BRCA tumors from EAs compared with AAs (Supplementary Fig. 6c and Supplementary Data 19), suggesting that population differences in APA could be driven by population differences in PA-related machinery gene expression.

We next asked what the underlying genetic basis behind these PSI differences could be. We firstly reasoned that if there is a genetic cause, it is likely to be somatic as we did not observe significant differences in APA in non-involved tissues between

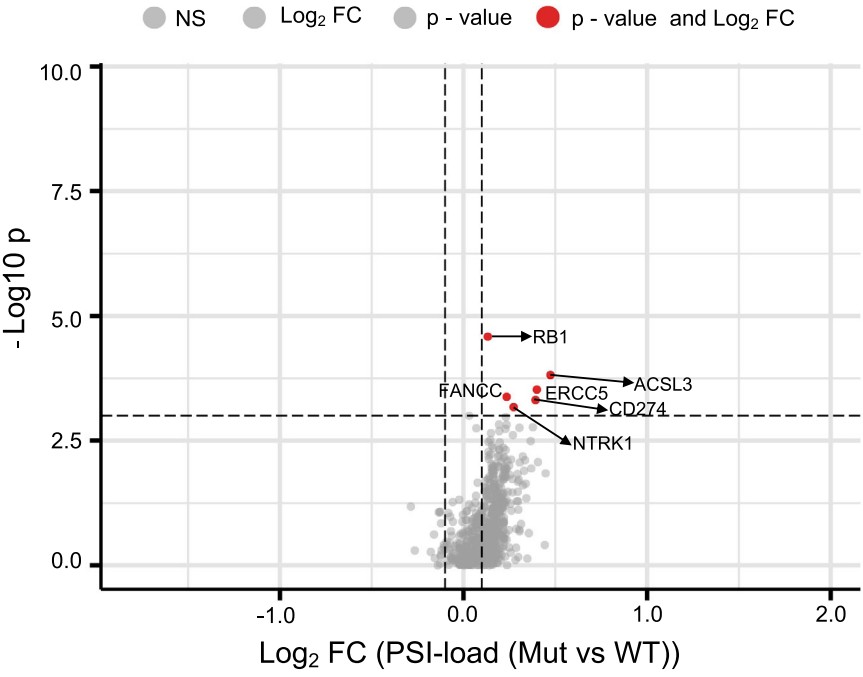

**Fig. 5 Ratio of 3′UTR shortening index (PSI index) in tumors with driver gene mutated vs. wildtype status from TCGA.** The ratio of median PSI-load index (fold change [FC], x axis) between driver gene mutated vs. wildtype tumors (pan-cancer) derived from TCGA for 683 cancer driver genes (COSMIC) and corresponding p-value of Wilcoxon rank-sum test are shown. The red points are significant (FDR < 0.1) past the FC threshold of FC > 0.1 and are labeled with their HGNC gene names.

EAs and AAs. Testing this hypothesis, we examined the relationship between somatic mutations in driver genes (682 driver genes census set derived from COSMIC [https://cancer.sanger.ac.uk/census]) with the 3′UTR shortening index in TCGA pan-cancer samples (Fig. 5 and Supplementary Data 20). The top genes whose mutation status are significantly associated (FDR  $p < 0.1$ ) with the 3′UTR shortening index after FDR correction are *RB1* (FC = 1.1, Wilcoxon rank-sum raw $p < 2.6E−05$), *ACSL3* (FC = 1.39, Wilcoxon rank-sum $p < 0.0001$), *ERCC5* (FC = 1.32, Wilcoxon rank-sum $p < 0.0002$), *PD-L1* (FC = 1.31, Wilcoxon rank-sum $p < 0.0004$), *FANCC* (FC = 1.17, Wilcoxon rank-sum $p < 0.0004$), and *NTRK1* (FC = 1.21, Wilcoxon rank-sum $p < 0.0006$). As the occurrence of these mutations is not shown to vary by population, we further reason that somatic mutations are not driving the population differences in APA that we describe. This observation is also consistent with Xia et al. [24].

## Discussion
Our study uses both a specific 3′UTR sequencing method that allows simultaneous genome-wide polyA site identification and an analysis pipeline tailored to detect significantly regulated APA sites in lung cancer. Charting the comparative regulation of polyA sites in lung cancer, we identify a global shortening of mRNA transcripts in lung tumor cells relative to non-transformed adjacent tissue. Almost 50% of genes with more than one polyA site undergo alternative polyadenylation in lung cancer to produce different transcript isoforms, implying a high degree of transcriptional heterogeneity at the level of the 3′UTR that is not usually captured in transcriptomic studies. The global and gene-specific (*DICER1, FGF2, TACC1, RAB10, RUNX1*[21], *ZEB1, PDXK, COL1A2, PPP3CB, C6orf89, SAP30L, APH1B, CHURC1*[31]) shortening observed in our study is consistent with previous findings in other cancer types using different approaches[21,31,46].

Pathway analyses suggest that cell proliferation and oncogenic signaling pathways, including mTOR, are enriched with short transcript isoforms in lung cancer. Intriguingly, the ubiquitin-mediated proteolysis pathway, a major target of mTORC1, is significantly enriched in the same direction. A recent study demonstrated that the activation of the mTORC1 pathway is a potential driver of global 3′UTR shortening in cancer[33]. Long isoforms are enriched with p53 signaling and metabolic-related pathways. These patterns are consistent with several previous observations[11,21,40,47–49]. Our results suggest that the generation of shorter 3′UTR transcripts through APA contributes to the molecular features of lung cancer, as evidenced by the grouping of genes undergoing APA in lung cancer-related pathways.

The effect of APA on gene expression, not yet fully understood, is debated and likely multi-modal. One hypothesis is that expression of a shorter 3′UTR leads to increased mRNA[50] or protein[19–21] *in cis*, a type of gene amplification in the absence of mutations or somatic alterations[49]. Our data indicate that genes with short mRNA transcripts have a tighter correlation with their respective protein products. This observation agrees with the decrease in concordance between mRNA and protein abundance in some tissues[51,52] and is consistent with studies showing a stronger gene-wise mRNA/protein correlation in tumor cells than in normal cells[47]. Others argue that the selection of short mRNA 3′UTRs through APA is enriched in transcripts predicted to act as ceRNAs for tumor-suppressor genes[14]. These associations are also complicated by the incomplete complementarity between miRNAs and mRNAs in mammals and the presence of other RNA stability regions in 3′UTRs, such as ALU elements. The latter also impacts the stability and expression of a mRNA and which, equally to miRNAs, can be impacted by 3′UTR shortening[53,54]. Further, as our data show, increased use of a proximal APA site in tumor need not necessarily lie in the 3′UTR and there are many instances of increased proximal APA use occurring in an intron or exon. This kind of APA may not impact

RNA or protein expression per se, but would almost surely impact the protein type produced. Thus, the ultimate impact of APA on the transcriptome and proteome is likely dependent on a multitude of factors.

Given recent evidence linking APA as a cellular stress response to exogenous stimuli[14,35,55], we hypothesized that smoking could be a potential global regulator of APA in lung cancer. Our data suggest that while smoking may contribute to the APA of a subset of genes, it does not appear to be a global regulator. This conclusion is consistent with other studies showing global shortening of 3′UTRs in cancers not associated with smoking[31].

The greater analysis depth of our sequencing method and study design facilitated the identification of APA events. A key finding from our study is that non-coding RNAs also undergo recurrent alternative polyadenylation in lung cancer and that it is a common event. Moreover, it seems to be a mechanism of long non-coding RNA expression, whereby the retention or loss of RNA via APA modulates binding by HuR, and likely, other RNA binding proteins as well. While previous work demonstrated that both long non-coding RNAs and microRNA transcripts undergo polyadenylation, and that alternative polyadenylation of lncRNA transcripts was previously described[22,46,56–58], our study demonstrates widespread APA of recurrent and clinically relevant non-coding RNAs in cancer and may represent a form of transcriptomic regulation for non-coding RNAs in cancer.

We also identified population differences in APA. On average, mRNA transcripts from EAs were two times more likely that AAs to undergo APA in lung cancer compared with non-involved tissues. However, we did not find a globally shorter 3′UTR landscape in lung cancer in either our own samples or in TCGA. As population differences in APA had not previously been addressed to our knowledge, we leveraged the TCGA database and observed significant global shortening of 3′UTRs among EAs with breast cancer compared with AAs and an inverse trend in esophageal cancer. We did not observe population differences in the use of specific polyadenylation signal hexamers among regulated PAS (Supplementary Fig. 6a). Previous studies have shown that most APA events result from the dysregulated expression of proteins controlling polyadenylation[24]. We and others[18] observed overexpression of key genes involved in the regulation of the polyadenylation (PA) machinery, including *CPSF* and *CSTF*, in tumors compared with non-involved lung tissues and we found a strong correlation between the expression of 3′end processing machinery genes and shorter 3′UTR length, consistent with the previous studies[18]. We, therefore, reasoned that the differences in APA between EAs and AAs could be due to global patterns of 3′end machinery expression or focal regulation by specific APA factors. In breast cancer, the occurrence of shorter 3′UTR transcripts was indeed significantly linked with a higher PA machinery score. Interestingly, Tang and colleagues recently demonstrated a higher concordance between mRNA and protein in breast tumors from AAs compared with EAs. Our findings in breast cancer are consistent with these findings, where in general, AAs had a longer 3′UTR compared with EAs, implicating tighter control and regulation of mRNA translation[47].

A key strength of this study is our use of a specific method to sequence and map polyA sites, rather than traditional 5′–3′ sequencing methods. This method circumvents the issue of saturation encountered by previous 5′–3′ RNA-seq methods and enables a comprehensive identification of APA events, including, for example, our identification of non-coding RNA isoforms and recurrent APA in lung cancer. Previous APA studies of cancer have used EST databases[59] and microarrays[11,18], which, though not incorrect, are limited by dependence on the annotation of APA sites by EST databases, the proportion of genes covered by the probes, probe design, and technical difficulties when a gene

has more than two polyA sites[10]. Methods based on traditional RNA-seq also have limitations. For example, while it is possible to detect polyA sites through the detection of sequences with stretches of As, only a small proportion of RNA-seq reads contain polyA tails, limiting the ability to identify APA events[26,27]. Of note, an ultra-deep sequencing study identified only ~40,000 putative polyA reads (~0.003%) from 1.2 billion total RNA-seq reads[25]. Also, short 3′UTRs are often embedded within long ones, and thus isoforms with short 3′UTRs are commonly overlooked by transcript assembly tools, though methods have been developed to overcome these limitations[24]. Thus, by leveraging 3′end-enriched RNA-seq methods, we have been able to comprehensively map the APA landscape in lung cancer.

Another strength of our study is the use of predominantly matched tumor and adjacent non-involved tissues and ERCC spike-in RNAs as controls for the sequencing method. It is possible that our study has missed some polyA sites—approximately 20% of human polyA sites do not contain a canonical hexamer (or one of its variants), instead relying on binding to the UGUA or downstream sequences[60]. Also, our analysis filtered out polyA sites without a well-characterized hexamer in the upstream pipeline making it possible that we overlooked a small portion of regulated APA events. A limitation of our work regarding population differences in APA that should be considered is our use of self-reported race in the classification of EAs and AAs. The race is a socially constructed variable that categorizes people based on non-biological characteristics.

Some, not all, members of the same race may also share common biology. We have previously shown that African American lung cancer patients have greater enrichment of genomic, transcriptomic, and proteomic profiles[28,61]. However, these biological determinants cannot account for all the differences observed. While we cannot pinpoint social, environmental, or genetic factors that would impact population differences in APA, in line with recent recommendations we have tried explaining how our populations were selected[62].

In summary, our study describes a level of transcriptomic regulation by APA that is associated with patient survival in lung cancer and yields insight into tumor biology and transcriptional heterogeneity that are missed by general transcriptome analyses. We find gene-level population differences in APA in lung cancer, and both gene-level and global differences in 3′UTR length in breast cancer. Finally, our description of recurrent and clinically relevant APA events in non-coding RNAs in lung cancer identifies a form of non-coding RNA expression control whereby APA acts as a dynamic docking location for RNA binding proteins.

## Methods

**Study subjects**. Ninety-eight patients with histologically confirmed non-small cell lung cancer were selected from the National Cancer Institute-Maryland (NCI-MD) lung cancer study. This NCI-MD Case Control Study was conducted in accordance with the Declaration of Helsinki. Institutional review board approval was granted from NCI and participating hospitals and registered on clinicaltrials.gov [https://clinicaltrials.gov/ct2/show/NCT00339859]. The cohort included 46 AAs and 52 EAs (Table 1). The population accrual and eligibility criteria for this study were previously described[63,64]. Written informed consent was obtained from each participant.

Individuals in this study self-reported as either African American or European American. Our previous work with this population indicates that the median West African ancestry in self-reported African Americans is 77%[61,65]. Eligible participants took part in a structured in-person interview from which demographic and lifestyle factors were documented. Clinical data were obtained from medical records and pathology reports for each patient. Fresh human lung tumor and matched adjacent non-involved lung tissues were obtained from patients directly after surgery. Each tissue was transferred to a sample collection tube, flash frozen, and stored at −80 °C until molecular analyses were performed. The study included 98 tumor tissues and 98 adjacent non-involved tissues ($n = 196$ total).

| Table 1 Demographic characteristics of the population. | | | |
|---|---|---|---|
| | **African American** | **European American** | *p*-value |
| Age | 63.7 (8.4) | 65.2 (9.3) | 0.16 |
| Sex | | | |
| Male | 10 (21.7) | 17 (32.7) | 0.28 |
| Female | 36 (78.3) | 35 (67.3) | |
| Smoking status | | | |
| Never | 3 (6.5) | 3 (5.8) | |
| Former | 15 (32.6) | 19 (36.5) | 0.78 |
| Current | 25 (54.3) | 29 (55.8) | |
| Missing | 3 (6.5) | 1 (1.9) | |
| Histology | | | |
| Adenocarcinoma | 25 (54.3) | 21 (37.5) | |
| Squamous | 14 (30.4) | 21 (37.5) | 0.57 |
| Other | 7 (15.2) | 10 (19.2) | |
| Stage | | | |
| I | 23 (50.0) | 35 (67.3) | |
| II | 14 (30.4) | 12 (23.1) | 0.27 |
| III | 7 (15.2) | 3 (5.8) | |
| Missing | 2 (4.4) | 2 (3.9) | |

**RNA library preparation and sequencing**. Construction of RNA libraries was performed using the QuantSeq Rev 3′mRNA-Seq Library Prep Kit for Illumina (Lexogen, Vienna, Austria) according to the manufacturer's instructions. Briefly, 500 ng of total RNA was reverse transcribed with an oligo(dT) primer containing an Illumina-specific linker sequence, followed by removal of RNA and second-strand cDNA synthesis with random primers. The resulting double-stranded cDNA was purified with magnetic beads. Libraries were amplified by PCR with primers containing Illumina adaptors and sample-specific barcodes and then purified with magnetic beads. The High Sensitivity DNA analysis Kit (Agilent, Santa Clara, CA) was used for library quantification.

Libraries were pooled and sequenced on the Illumina Hiseq 2500 with 100 bp single-end reads. Eight samples were run per lane, including 4 tumors and 4 adjacent non-involved samples derived from 2 AA and 2 EA to limit bias. The QuantSeq Rev kit includes a customized sequencing primer that anneals to the linker sequence previously introduced in the oligo(dT) priming step. Reads are strand-specific and generated from the 3′untranslated region and last exon in a 3′–5′ direction. The method generates one fragment per transcript, which also facilitates gene expression analyses. In total, we sequenced 196 samples, which produced, on average, 20.6 million reads per sample.

**Generation of the global polyadenylation atlas**. Data analysis was performed with expressRNA[66] (expressRNA.org; https://github.com/grexor/apa), an open-source modular and scalable platform that provides a complete analytical framework incorporating tools for read alignment, genome annotation, and calling of differentially polyadenylated genes.

The 3′UTR sequence data were processed and mapped to the genome using STAR, which allows soft clipping from both 5′ and 3′ ends. On average, 74% of the reads from each sample aligned to the hg38 reference genome. We then considered the locus of the last nucleotide of each alignment at the 3′ end and clustered the signal across all samples to obtain a global polyadenylation atlas. Since oligo-dT priming is known to cause internal-priming events, we only considered loci that had any of the 18 characterized PAS in the [−30….−10] nt region relative to the detected cleavage site[30]. We additionally filtered out genomic positions that had more than 6 consecutive As or more than 8 sparse As in any 10 nt window in the regions spanning the detected cleavage site ([−10….10] nt) to filter out internal-priming events.

Since cleavage is not a nucleotide exact process and tends to occur within a small window of positions around the dominant cleavage site, we summed the read counts from cleavage sites up to 5 nt away from each dominant polyA site[66]. We also selected a representative cleavage site for each of the PAS signals. To focus on fully independent cleavage sites with their own PAS and cis-regulatory elements, we identified the dominant polyA sites based on read count and ranked them in descending order, such that all resulting sites were at least 125 nt apart[66]. Since 3′ UTR isoforms are not always fully annotated, we added 5 kb of the intergenic region downstream of each gene. If two genes were closer than 10 kb, only the region up to the middle of the beginning of the downstream gene was added. To avoid spurious cleavage and polyadenylation events, we only kept polyA sites that were present across at least 25% of samples[66]. To avoid poorly used sites, polyA sites with less than ten read counts in either adjacent non-involved or tumor tissues were filtered out. We then discarded those polyA sites that had less than 5% of reads compared with the major polyA site in the same gene[66].

We further classified regulated polyA sites based on their position in the gene; (a) same-exon pairs—if both proximal and distal sites are located in the same exon, (b) composite-exon—if the proximal site is part of a composite-exon, i.e., containing an internal 5′ splice and the two sites were generated by alternative use of that splice site, and (c) skipped-exon sites—if the proximal site is part of an exon that is fully spliced[66].

**Correlation between non-coding RNA expression and APA**. We first performed quality and adapter trimming by using the cutadapt tool to remove the NEB 5′ (GTTCAGAGTTCTACAGTCCGACGATC) and 3′ (AGATCGGAAGAGCACA CGTCT) adapters. For isomiRs, we mapped reads to precursors from miRbase version 22 using exhaustive string comparisons while disallowing mismatches, insertions, or deletions. IsomiRs were then annotated to include offset positions[67] to indicate how their 5′ and 3′ ends differ from that of the annotated reference mature.

We used available microRNA binding site annotation from APADB to enrich the polyA sites. Bedtools set operation were used to assess the lost microRNA binding sites. Correlation was carried out by using the log2 tumor and normal PSI ratio and gene transcripts per million (*TPM*) from RSEM for 60 LincRNA matched genes. For LincRNA isoforms, the identification and quantification of APA events between tumor and matched normal tissues was performed by using delta Percentage of Distal polyA site Usage Index (dPDUI or ΔPDUI) from DaPARS[24] with all default parameters. The same trend was defined as that shorten 3′ APA corresponds with lower gene expression and genes on opposite trend as that shorten 3′ APA corresponds with higher gene expression in tumor and normal samples.

**Derivation of PSI from TCGA samples**. The ΔPDUI measure derived from DaPARS[24] were mined from The Cancer 3′ UTR Atlas (TC3A), a comprehensive resource of APA usage for 10,537 tumors across 32 cancer types http://tc3a.org and converted to PSI (1- PDUI). Codes generated to analyze this data set are provided at https://github.com/ruppinlab/apa_lung.git. Race information is self-reported.

*CTPAC analysis*: We downloaded the protein abundance information for primary breast cancer samples of the TCGA cohort from CTPAC portal [https:// cptac-data-portal.georgetown.edu/datasets] and downloaded their respective mRNA expression profile from GDAC [https://portal.gdc.cancer.gov/] firehose portal. We next ranked each gene by its median PSI index higher to lower and asked whether the top 10% genes going (high 3′ shortening in tumors) have a stronger correlation between their mRNA and protein across patients than the bottom 10% of the genes (high 3′ elongation in tumors). We repeated this analysis for lung tumors samples by mining the mRNA-protein correlation information from Gillette et al.[41]. Codes used to analyze this data set are provided at https:// github.com/ruppinlab/apa_lung.git. Race information is self-reported.

**Analysis of RIP-seq data**. We downloaded the RNA immunoprecipitation (RIP-seq) assays profile measuring ELAV1 DNA binding of transformed K562 leukemia cells and normal Gm12878 cells as a control from GSE35585 For a given gene or element (miRNA or long non-coding RNA), ELAV1 binding is quantified by counting the number of reads between primary proximal and distal polyA site of usage in both cell lines individually. Codes used to analyze this data set are provided at https://github.com/ruppinlab/apa_lung.git. Race information for this data set is self-reported.

**Calculation of Polyadenylation Machinery Index**. We mined the expression levels of 19,001 genes in 8749 tumor samples from TCGA cohort[68]. For a subset of core regulatory genes of polyadenylation of mRNA pathway (biocarta_cpsf_pathway, M22041, [https://data.broadinstitute.org/gsea-msigdb/msigdb/biocarta/human/h_cpsfPathway.gif]), we identified their median *z*-score expression for each sample and defined it as PA machinery activity index. Codes used to analyze this data set are provided at https://github.com/ruppinlab/apa_lung.git. Race is self-reported for this data set.

**Identification of clinically relevant APA events**. To determine whether APA captures genes with clinical relevance, we first performed a lasso regression (feature selection step) for each gene, which yielded 21 genes whose PSI values are associated independently with survival. Using the PSI levels of these 21 genes and their corresponding hazard ratio from multivariate regression correcting for patient age, sex, race, and tumor stage as their linear weights, we generated a prognostic index for each sample in leave-one-out cross-validation. Thus, the Prognosis Index (PI) of *k*th sample is defined as,

$$\text{PI} = \sum_{i=1}^{m} \beta i \chi i$$

where, $\beta$ is the cox-risk coefficient of gene $i$, $\chi$ is the gene $i$ PSI value. The patients with a PI greater than the median are categorized as low-risk and the rest as high-risk (Fig. 2f).

**Reporting summary**. Further information on research design is available in the Nature Research Reporting Summary linked to this article.

## Data availability

The NCI-MD data were derived from patients enrolled in the ongoing NCI-MD Case-Control Study [https://clinicaltrials.gov/ct2/show/NCT00339859]. The fastq files, race and processed data of the data set generated during the current study have been uploaded to GEO repository under accession number GSE174330. The processed data are available in the Supplementary Data file. PolyA database in human tissues published by Derti et al.[29], is available from GEO repository under accession number GSE30198. PolyA database in human tissues published by Gruber et al.[69], is available from NCBI Read Archive under accession number SRP065825. The TGCA data used in this study can be accessed at http://tc3a.org. The RIP-Seq data used in this study are available from GEO repository under accession number GSE35585. The CTPAC data portal is accessible at https://cptac-data-portal.georgetown.edu/datasets. The GDAC data can be accessed at https://portal.gdc.cancer.gov/. Raw data for all the remaining figures are provided in the Supplementary Data file. Source data are provided with this paper.

## Code availability

Codes generated in this study are provided as a Supplementary Software file.

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

## Acknowledgements

We sincerely thank Dr. Gregor Rot, University of Zurich, Switzerland for his assistance with the expressRNA analysis. This work was supported by the Intramural Research Program of the Center for Cancer Research, National Cancer Institute.

## Author contributions

B.M.R. and A.Z. designed the research. A.Z., K.A.M., and E.D.B. performed experiments. S.S., M.A., C.N., N.S., Q.C., C.Y., P.L., D.M., E.R., and A.Z. analyzed the data. B.M.R. and A.Z. wrote the manuscript. A.Z., S.S., M.A., C.N., D.D., E.D.B., N.S., K.A.M., Q.C., C.Y., P.L., D.M., E.R., and B.M.R. reviewed and edited the manuscript.

## Competing interests

The authors declare no competing interests.
