## [Peer Review File · Nature Communications]

REVIEWER COMMENTS

Reviewer #1 (Remarks to the Author): Expert in polyadenylation

Review summary:

This manuscript from Zingone et al. analyzed alternative polyadenylation (APA) genome-wide in lung cancer cells. They compared samples from European American and African American populations. They found global transcript shortening due to APA, in both mRNAs and non-coding RNAs. They found that 3'UTR shortening correlates with poor cancer survival and can make mRNA and protein levels more consistently related. Overall, this is a comprehensive work with some interesting messages. However, the data are largely descriptive and major conclusions are confirmatory of previous reports.

Major:

--the difference between EA and AA samples is interesting. But what does not mean functionally? Any genetic reasons for the difference?

--there are several cancer subtypes, such as adenocarcinoma, large cell carcinoma, squamous cell carcinoma, etc. Do they all follow the same trend in APA? How about their difference between AA and EA?

--why 3'UTR shortening makes mRNA and protein levels more correlated. Is this because of expression levels? Note that highly expressed genes tend to have a better concordance between mRNA and protein levels.

Minor:

--The manuscript contains many errors that require a careful check. Some are shown below.

--title: two "in".

--"The greater resolution provided by this 3' to 5' sequencing method has **not** enabled us to identify new gene-level APA events in lung cancer, including the discovery of many recurrent APA events in long non-coding RNAs and miRNAs genes, as well as population differences."

-- The low or high risk in figure 2F means 3'UTR shortening or lengthening?

-- In figure 3A, how to explain the miR17HG, with lower PSI, which means longer 3'UTR, yet produces more mature miRNAs?

--ELAV1/HuR **RNA** binding in figure 3C.

--In Figure 4C, Blue and Red represented EA or AA, respectively?

--In supplementary table 11, how to explain the negative correlation between PSI and expression?

Reviewer #2 (Remarks to the Author): Expert in bioinformatics

Zingone et al. used a 3'UTR specific sequencing method to directly investigate the landscape of alternative polyadenylation (APA) and its role in lung cancer using a racially balanced dataset. The major novelty of this study is that the authors applied Quantseq to avoid the saturation problem in 5'-3' RNA-seq data, thus allowing them to identify more dynamic APA events. The author also identified APA differences between European America and African America. I have the following

concerns:

Major concerns:

1. In Figure 2E, the figures show pathways that are not significantly enriched with $p\text{-value} < 0.05$ as the author claimed in the legend. If the $p\text{-value} < 0.05$, the $-\log_{10}(P) > 1.3$ and clearly, a lot the pathways showed in Figure 2E do not meet the criterion. What's more, the axis looks strange for longer transcripts plot (the one in blue), distance between 0 and 2.0 is significantly less than distance between 2.0 to 2.5.
2. In Figure 2F, can the authors explain whether the survival plot is based on APA of a single gene (if so, which gene) or based on APA of the 12 identified prognostic genes combined?
3. Can the authors give some example scatterplots of gene-wise correlation between PSI and expression? The Spearman's coefficient is negative means PSI increases when gene expression decreases, which seems to contradict the existing knowledge that 3'UTR shortening (larger PSI) is associated higher abundance of mRNA (higher expression). Could the author better explain the relationships?
4. Can the authors better explain the reasons why they perform the steps in line 224-line 226?
5. I am not sure why the authors include supplementary Fig. 5A (BRCA dataset) since the manuscript is about lung cancer.
6. Can the authors explain the scatterplot in Figure 3A? what does the dots represent? (if it represents host miRNAs, then isn't there supposed to be 8 points instead of 10?) In addition, what is the correlation coefficient and p-value?
7. Similar concerns with Figure 3B.
8. Can the author explain what is the shortening in EA and AAs referring to in TCGA 5' -3' RNA-seq from non-small cell lung cancer? What are the genes in Tumors being compared to since adjacent non-involved samples are not included in this part of the study?

Minor concerns:

1. This paper still needs some proofreading. For example, the author put an extra "in" in the title. In addition, in line 110, the author mistakenly added a "not" to their main conclusion...
2. The figures need to be in clear format, some of the figures are very blur in this version of the manuscript.

Reviewer #3 (Remarks to the Author): Expert in health disparities and genetics

In this manuscript, the authors explore the functional role of alternative polyadenylation in determining transcriptome heterogeneity in lung cancer. I have reviewed the manuscript and have the following comments regarding their population:

- 1) The authors have entitled their manuscript "A comprehensive map..... in a racially balanced African American and European America lung cancer cohort". It is unclear what the authors mean by "racially balanced". The inclusion of both African descent and European descent samples does not make this a "racially-balanced" study as the US population consist of many other race/ethnic groups – this study includes only a small sample from two race/ethnic groups. Therefore, I would advise that the authors remove the term "racially-balanced" from their title and abstract.

2) In addition, the authors should indicate that this is a case study and that the subjects were derived from a case-control study as the authors have stated in line 572 - 573. The use of the word "cohort" in the title can be misleading.

3) The authors also do not specify whether the samples they used in their study are self-identified as African American or European American or if they (the investigators) have ancestry informative markers (AIMs) on these samples. It would be advisable that the authors make clear that they do have AIMs on their samples in their method section. If not, the authors should indicate that these samples are from subjects who self-identified as belonging to one race/ethnic group. Lack of AIMs will weaken the results that looked at differences in alternative polyadenylation by ancestry.

4) The authors should note that race is a social construct and use of the word race within the context of this study is not appropriate. The authors should change "race" to "ancestry" in lines 332, 336 (and any other areas where "race" was used).

5) The demographic and relevant clinical information on the 96 subjects should be included in the manuscript or supplemental tables. The authors indicated that it was present in "Table 1" in line 465 in "Material and Methods" section but I do not see it anywhere in the manuscript or supplementary tables excel file.

Reviewer #4 (Remarks to the Author): Expert in health disparities and genetics

Major comments:

1. Figure 2F: Details of the association between APA events and survival are unclear. How was low risk and high risk defined for the APA events? For the Kaplan Meier curves, how was the cut-point of high vs low risk determined? It is unclear how the APA pattern for the 12 genes was modeled. Was a 12-gene score developed to summarize all genes? What p-value threshold was used to determine the 12 genes to include? Supp Table 7 appears to have 13 significant genes ($p < 0.05$). Given that the number of genes undergoing shortening is greater in EA than AA, should a different set of genes comprise the 12-gene panel for EA and AA?

2. Clinical features: It appears neither treatment nor race were included in the Cox regression model. Given the known impact of treatment on lung cancer survival, racial differences in lung cancer survival, and the authors' demonstration of racial differences in APA events, these are likely important variables to consider. How does the lack of including covariates race and treatment in the Cox model impact the authors' findings? Does the APA association with survival hold in both race groups?

Minor comments:

1. Table 1 is referenced in the manuscript but appears to be missing from the files.

2. Figure 2C is mentioned in the text after Figure 2D.

3. Whether race was self-reported should be stated in the text. While the authors cite their prior work with descriptive characteristics of the study population, it is appropriate to provide to the

reader information details as to how race was ascertained given the manuscript emphasis on racial differences.

4. Figure 4C. X-axis shows genes yet appears to be mislabeled as Race.

5. Page 4, line 110. Extra word 'not' appears and should be removed, i.e. "The greater resolution provided by this 3' to 5' sequencing method has not enabled..."

6. Page 10, line 226. Text indicates greater or equal to 1,156. I think the authors are referring to a critical value of 1.156. Change comma to decimal.

Response to Reviewers

Review summary

This manuscript from Zingone et al. analyzed alternative polyadenylation (APA) genome-wide in lung cancer cells. They compared samples from European American and African American populations. They found global transcript shortening due to APA, in both mRNAs and non-coding RNAs. They found that 3'UTR shortening correlates with poor cancer survival and can make mRNA and protein levels more consistently related. Overall, this is a comprehensive work with some interesting messages. However, the data are largely descriptive and major conclusions are confirmatory of previous reports.

Major

1) the difference between EA and AA samples is interesting. But what does not mean functionally? Any genetic reasons for the difference?

We agree with the reviewer that this is an interesting observation. In terms of the hypothesis that led us to ask this question in the first place, we did anticipate that there could be more 3'UTR shortening among EAs, given that our previous work¹ indicated that although there is a good deal of consistency in the transcriptome between EAs and AAs, of the parts that differed, we saw an increased representation of cell proliferation pathways among EAs. As proliferating cells have more 3'UTR shortening^{2,3}, our observations were consistent with our initial hypothesis.

As to what they mean functionally, there are a few possibilities; one of which is that the genes perturbed by APA in EAs and not AAs could give rise to more protein (through loss of miRNA or ALU regulation³). Alternatively, in line with the competing endogenous RNA hypothesis⁴, the loss of a pool of miRNA binding sites on one gene's 3'UTR is likely to modulate the probability of the cognate miRNA binding to the 3'UTR of other genes in the network.

Addressing the second question, we first reasoned that if there is a genetic cause, it is likely to be somatic as we did not observe significant differences in APA in non-involved tissues between EAs and AAs. Testing this hypothesis, we examined the relationship between somatic mutations of each driver gene (683 driver genes census set derived from COSMIC; <https://cancer.sanger.ac.uk/census>) with the 3'UTR shortening index in TCGA (below and revised manuscript Supp Table 19). The top genes whose mutation status are significantly associated (FDR $P < 0.1$) with 3'UTR shortening index after FDR correction are *RBI* (FC=1.1, Wilcoxon rank sum raw $P < 2.6E-05$), *ACSL3* (FC=1.39, Wilcoxon rank sum raw $P < 0.0001$), *ERCC5* (FC=1.32, Wilcoxon rank sum raw $P < 0.0002$), *PD-L1* (FC=1.31, Wilcoxon rank sum raw $P < 0.0004$), *FANCC* (FC=1.17, Wilcoxon rank sum raw $P < 0.0004$), and *NTRK1* (FC=1.21, Wilcoxon rank sum raw $P < 0.0006$). As the occurrence of these mutations is not shown to vary by population^{4,5}, we therefore concluded that somatic mutations are not driving the population differences in APA that we describe. This observation is also consistent with Xia et al.⁶

Revision in text:

Page 13: We next asked what the underlying genetic basis behind these PSI differences could be. We firstly reasoned that if there is a genetic cause, it is likely to be somatic as we did not observe

significant differences in APA in non-involved tissues between EAs and AAs. Testing this hypothesis, we examined the relationship between somatic mutations in driver genes (683 driver genes census set derived from COSMIC; <https://cancer.sanger.ac.uk/census>) with the 3'UTR shortening index in TCGA pan-cancer samples (Figure 5, Supp Table 19). The top genes whose mutation status are significantly associated (FDR $P < 0.1$) with the 3'UTR shortening index after FDR correction are *RB1* (FC=1.1, Wilcoxon rank-sum raw $P < 2.6E-05$), *ACSL3* (FC=1.39, Wilcoxon rank-sum $P < 0.0001$), *ERCC5* (FC=1.32, Wilcoxon rank-sum $P < 0.0002$), *PD-L1* (FC=1.31, Wilcoxon rank-sum $P < 0.0004$), *FANCC* (FC=1.17, Wilcoxon rank-sum $P < 0.0004$), and *NTRK1* (FC=1.21, Wilcoxon rank-sum $P < 0.0006$). As the occurrence of these mutations is not shown to vary by population, we further reason that somatic mutations are not driving the population differences in APA that we describe. This observation is also consistent with Xia *et al.*⁶

Figure 5: Ratio of 3'UTR shortening index (PSI index) in tumors with driver gene mutated versus wildtype status from TCGA. The ratio of median PSI-load index (fold change [FC], x-axis) between driver gene mutated vs. wildtype tumors (pan-cancer) derived from TCGA for 683 cancer driver genes (COSMIC) and corresponding P -value of Wilcoxon rank-sum test are shown. The red points are significant (FDR <0.1) past the FC threshold of FC >0.1 and are labeled with their HGNC gene names.

2) there are several cancer subtypes, such as adenocarcinoma, large cell carcinoma, squamous cell carcinoma, etc. Do they all follow the same trend in APA? How about their difference between AA and EA?

We appreciate the reviewer's consideration here of the multiple layers of lung cancer histology. When we compared differences in APA between LUAD and LUSC we did not see meaningful differences (Supplementary Figure 2B) (Wilcoxon rank-sum $P < 1$). We had then computed the genes that are shortened in LUSC (N=2,404) and LUAD (N=2,740) separately and observed very strong enrichment between these two sets (Figure 1C, Jaccard Index of overlap=0.56, hypergeometric enrichment $P < 3E-41$), reflecting a similar landscape of APA between LUAD and LUSC. Our study included too large cell carcinomas, or other lung subtypes, to facilitate any analyses that were sufficiently powered.

With regard to testing whether there were population differences in each histology, we have limited statistical power due to the low number of samples that arise following sub-stratification. However, we repeated the histology analysis by population and did not find statistically significant differences in median 3'UTR length between AAs and EAs in either LUAD (Wilcoxon rank-sum $P < 0.34$) or LUSC (Wilcoxon rank-sum $P < 0.63$). Further, when we compared differences by EA and AA in LUAD and LUSC, we didn't see any significant differences. We have added this result as Supplementary Figure 2D and suggest that this needs to be further investigated with a larger sample size.

Revisions in text

Page 12: We repeated this analysis, comparing the gene-wise number of APA events (log2PSI value) in tumor vs non-involved adjacent tissues in both EAs and AAs using a multivariate regression correcting for patient age, sex, tumor stage and smoking status and consistently found a 2-fold higher number of genes going through APA events in EAs vs AAs. **There was no specific enrichment in LUSC vs LUAD (Supplementary Figure 2d).**

Supplementary Figure 2: Distribution of polyA site index (PSI) in tumor and normal samples. **a** Higher PSI indicates shorter 3'UTR. **b** Shared and distinct APA events in tumor vs. non-involved adjacent samples between LUAD and LUSC subtypes. **c** The median 3'UTR length (tumor/normal) between LUAD and LUSC tumors samples. **d** The median 3'UTR length (tumor/normal) in AA and EA for both LUAD or LUSC samples separately.

3) why 3'UTR shortening makes mRNA and protein levels more correlated. Is this because of expression levels? Note that highly expressed genes tend to have a better concordance between mRNA and protein levels.

This is a good question and we thank the reviewer for raising it. Shortening of 3'UTR can lead to loss of miRNA binding sites (the latter, when engaged by miRNA transcripts can induce

message degradation or destabilization) or ALU elements, for example. Escape from this miRNA-induced degradation can result in a more stable transcript with longer half-life and thus a higher protein translation rate^{2,3}. This, therefore, can lead to a stronger correlation between protein abundance and mRNA abundance. We have expanded upon this reasoning in the “Results” section.

Revision in text

Page 8: **When a miRNA engages with its cognate miRNA binding site, it can induce message degradation or destabilization. Thus, escape from miRNA repression can result in a more stable transcript with longer half-life and relative higher abundance of the mRNA and resulting in more protein^{2,3}. Alternatively, mRNA levels are not always affected, given that most miRNA/mRNA binding interactions are imperfect and do not lead to mRNA degradation. Lastly, the ceRNA hypothesis argues that, rather than affecting mRNA expression *in cis*, APA impacts mRNA *in trans* by altering the stochastic nature by which the loss of one miRNA binding site frees up miRNAs to bind to other targets^{2,7,8,9,10,11}. To test these hypotheses in lung cancer, we first found that among genes with significant 3’UTR shortening, there was an average loss of nine miRNA binding sites per gene (range 1-51), with enrichment for lung cancer-associated miR-124, miR-181, let-7 and miR-27 binding sites (Supplementary Figure 4A & 4B) (Supplementary Table 10 & 11). We calculated a gene-wise correlation between PSI and expression and found that the PSI of 1,376 of the 3,351 genes significantly correlates with expression (Supplementary Table 12), roughly evenly split between positive and negative associations. We then asked, what is the probability of observing this number of genes (N=1,376) or greater by chance. To test this, we shuffled the APA matrix 10,000 times and calculated an empirical significance ($P < 0.17$) by counting the number of times the genes with Spearman Rho > 0.1 are greater than or equal to 1,376.**

Minor

1) The manuscript contains many errors that require a careful check. Some are shown below.

--title: two “in”.

--“The greater resolution provided by this 3’ to 5’ sequencing method has *not* enabled us to identify new gene-level APA events in lung cancer, including the discovery of many recurrent APA events in long non-coding RNAs and miRNAs genes, as well as population differences.”

We apologize for these errors and have now carefully proofread the manuscript to rectify all possible errors, including those pointed out by the reviewer.

2) The low or high risk in figure 2f means 3’UTR shortening or lengthening?

We are happy to clarify this point and would also like to first correct a typo in this section, where the number of genes used in the prognostic signature is 21 and not 12.

To determine whether APA captures genes with clinical relevance, we first performed a LASSO regression (feature selection step) for each gene, which yielded 21 genes whose PSI

values are associated independently with survival (provided in revised Supp Table 7). Using the PSI levels of these 21 genes and their corresponding hazard ratio from multivariate regression correcting for patient age, sex, ancestry and tumor stage as their linear weights (described in Methods and Supplementary Table 7), we generated a prognostic index for each sample in a leave-one-out cross validation. Thus, the Prognosis Index (PI) of k -th sample is defined as,

$$PI = \sum_{i=1}^m \beta_i \chi_i$$

where, β is the cox-risk coefficient of gene i , χ is the gene i PSI value. The patients with a PI greater than the median are categorized as low-risk and the rest as high-risk (Figure 2F). We observed that the patients in the high-risk group have a significantly higher lower survival than patients in low-risk (Log Rank $P < 0.001$). We have added a description of these steps used in the “**Identification of clinically relevant APA events**” section in Methods.

Revision in text

Page 24: **Identification of clinically relevant APA events**

To determine whether APA captures genes with clinical relevance, we first performed a lasso regression (feature selection step) for each gene, which yielded 21 genes whose PSI values are associated independently with survival. Using the PSI levels of these 21 genes and their corresponding hazard ratio from multivariate regression correcting for patient age, sex, ancestry and tumor stage as their linear weights, we generated a prognostic index for each sample in a leave-one-out cross validation. Thus, the Prognosis Index (PI) of k -th sample is defined as,

$$PI = \sum_{i=1}^m \beta_i \chi_i$$

where, β is the cox-risk coefficient of gene i , χ is the gene i PSI value. The patients with a PI greater than the median are categorized as low-risk and the rest as high-risk (Figure 2f).

3) In figure 3A, how to explain the miR17HG, with lower PSI, which means longer 3'UTR, yet produces more mature miRNAs?

In the context of APA within non-coding RNAs, including miR-17HG, the consequence of APA can depend on whether the piece of RNA that is modulated by APA includes regions that have binding sites for stabilizing or destabilization by relevant proteins. In the instance of miR-17HG, there is increased use of a distal polyA site at chr13:91354575. The inclusion of this region in lung tumors—deduced from evidence of a lower PSI in tumor—and the production therefore of a longer transcript in tumors relative to normal, leads to increased available binding sites for the HuR proteins. As the PSI correlates with increased miR-17 cluster mature miRNA expression, we speculate that this HuR binding leads to greater transcript stabilization. As this is the first time that this observation has been described, we understand that further work is needed to increasingly elucidate the importance of this novel mechanism of miRNA expression in both normal physiology and cancer.

We appreciate this Reviewer’s perspective and have added greater context of this example in the text of the manuscript.

Revision in text:

Page 11: For example, increased use of a distal polyA site on the miR17HG RNA transcript relative to a proximal one (Supplementary Table 4) leads to the inclusion of a region with multiple binding sites for HuR proteins in tumor tissues (Figure 3a). Indeed, our data show that there is more HuR binding to this region in cancer, compared with normal, cell lines (Figure 3C). As the PSI correlates with increased miR-17 cluster expression, we speculate that this HuR binding leads to greater transcript stabilization, greater processing and, therefore, higher expression.

4) ELAV1/HuR *RNA* binding in figure 3C.

We thank the reviewer for catching this and have changed the legend to reflect RNA.

5) In Figure 4c, Blue and Red represented EA or AA, respectively?

Thank you for point this out, the missing legends have now been added.

6) In supplementary table 11, how to explain the negative correlation between PSI and expression?

The reviewer raises an important point regarding the relationship between the mRNA expression and PSI correlations. Across the 3,531 genes for which we observed a relationship between APA and tumor/normal status, we observed either strong, moderate or no correlation in both a positive and negative direction. The histogram of this correlation coefficient is provided below.

When a miRNA engages with its cognate miRNA binding site, it can induce message degradation or destabilization. Thus, escape from miRNA repression *can* result in a more stable transcript with longer half-life and relative higher abundance of the mRNA, resulting in more protein^{2,3}. Alternatively, mRNA levels are not always affected given that most miRNA/mRNA binding interactions are imperfect and do not lead to mRNA degradation (unlike in plants where the perfect complementarity leads to message degradation). Other studies have shown how loss or mutation of a miRNA binding site, while not affecting mRNA, can affect protein. Lastly, the ceRNA hypothesis argues that, rather than affecting mRNA expression *in cis*, APA impacts

mRNA *in trans* by altering the stochastic nature by which the loss of one miRNA binding site frees up miRNAs to bind to other targets^{2, 7, 8, 9, 10, 11}. Overall, our data reflect the myriad of ways in which 3'UTR shortening can affect mRNA levels and how that there isn't one way by which 3'UTR shortening can impact mRNA and protein. Indeed, as mentioned, one additional factor driving stronger protein and mRNA correlations is the abundance by which a protein is expressed. Our manuscript also excluded reference to ALU regions embedded within 3'UTRs, which also impact the stability and expression of a mRNA and which, equally to miRNAs, can be impacted by 3'UTR shortening^{12, 13}. We have tried to expand on these themes in the revised version of the manuscript to give the reader a fuller, and hopefully clearer, view of our findings.

Revision in text:

Page 8: When a miRNA engages with its cognate miRNA binding site, it can induce message degradation or destabilization. Thus, escape from miRNA repression can result in a more stable transcript with longer half-life and relative higher abundance of the mRNA and resulting in more protein^{2, 3}. Alternatively, mRNA levels are not always affected, given that most miRNA/mRNA binding interactions are imperfect and do not lead to mRNA degradation. Lastly, the competing-endogenous (ceRNA) hypothesis argues that, rather than affecting mRNA expression *in cis*, APA impacts mRNA *in trans* by altering the stochastic nature by which the loss of one miRNA binding site frees up miRNAs to bind to other targets^{2, 7, 8, 9, 10, 11}. To test these hypotheses in lung cancer, we first found that among genes with significant 3'UTR shortening, there was an average loss of nine miRNA binding sites per gene (range 1-51), with enrichment for lung cancer-associated miR-124, miR-181, let-7 and miR-27 binding sites (Supplementary Figure 4a & 4b) (Supplementary Table 10 & 11). We calculated a gene-wise correlation between PSI and expression and found that the PSI of 1,376 of the 3,351 genes significantly correlates with expression (Supplementary Table 12), roughly evenly split between positive and negative associations. We then asked, what is the probability of observing this number of genes (N=1,376) or greater by chance. To test this, we shuffled the APA matrix 10,000 times and calculated an empirical significance ($P < 0.17$) by counting the number of times the genes with Spearman Rho > 0.1 are greater than or equal to 1,376.

Page 15: The effect of APA on gene expression, not yet fully understood, is debated and likely multi-modal. One hypothesis is that expression of a shorter 3'UTR leads to increased mRNA¹⁴ or protein^{2,9,10} *in cis*, a type of gene amplification in the absence of mutations or somatic alterations¹⁵. Our data indicate that genes with short mRNA transcripts have a tighter correlation with their respective protein products. This observation agrees with the decrease in concordance between mRNA and protein abundance in some tissues^{16,17} and is consistent with studies showing a stronger gene-wise mRNA/protein correlation in tumor cells than in normal cells¹⁸. Others argue that the selection of short mRNA 3'UTRs through APA is enriched in transcripts predicted to act as competing-endogenous RNAs (ceRNAs) for tumor-suppressor genes¹¹. **These associations are also complicated by the incomplete complementarity between miRNAs and mRNAs in mammals and the presence of other RNA stability regions in 3'UTRs, such as ALU elements. The latter also impact the stability and expression of a mRNA and which, equally to miRNAs, can be impacted by 3'UTR shortening^{12,13}. Further, as our data show, increased use of a proximal APA site in tumor need not necessarily lie in the 3'UTR and there are many instances of increased proximal APA use occurring in an intron or exon. This kind of APA may not impact RNA or protein expression *per se*, but would almost surely impact the type protein produced. Thus, the ultimate impact of APA on the transcriptome and proteome is likely dependent on a multitude of factors.**

Reviewer #2 (Remarks to the Author): Expert in bioinformatics

Zingone et al. used a 3'UTR specific sequencing method to directly investigate the landscape of alternative polyadenylation (APA) and its role in lung cancer using a racially balanced dataset. The major novelty of this study is that the authors applied Quantseq to avoid the saturation problem in 5'-3' RNA-seq data, thus allowing them to identify more dynamic APA events. The author also identified APA differences between European America and African America. I have the following concerns:

Major concerns

1) In Figure 2E, the figures show pathways that are not significantly enriched with p-value < 0.05 as the author claimed in the legend. If the p-value < 0.05, the $-\log_{10}(P) > 1.3$ and clearly, a lot the pathways showed in Figure 2E do not meet the criterion. What's more, the axis looks strange for longer transcripts plot (the one in blue), distance between 0 and 2.0 is significantly less than distance between 2.0 to 2.5.

We thank the reviewer for pointing this out and agree with the comment. The IPA analysis applies a $-\log_{10}$ (p-value) cut-off of 1.3 ($P \leq 0.05$). Each pathway with a P -value equal or greater than 1.3 is then considered as statistically significant. In the initial version of Figure 2e, we had included some pathways that, though not statistically significant, seemed interesting and relevant to cancer biology. However, as mentioned, we agree with the reviewer and have therefore updated Figure 2E to include only the pathways that were significantly enriched. We have also updated the legend for the plot relative to the longer transcripts and apologize for any confusion. We have also added a full list of the IPA results as Supplementary Table 7.

Revisions in text:

Page 30: **Figure 2: Role of alternative polyadenylation in shaping the molecular and clinical features of lung cancer.** **a** Global shortening of 3'UTR detected by DexSeq2 analysis. The dot plot map shows genes undergoing 3'UTR shortening (red) or lengthening (blue) and no changes (gray) in tumors compared with adjacent non-involved tissues. **b** Genomic location of regulated proximal and distal sites in tumors. **c** Expression of polyadenylation-related genes in tumor and adjacent non-involved tissues. Heatmap of combined z-score and gene expression analysis of 29 known polyadenylation factors in tumors compared to non-involved tissues. The asterisk indicates the degree of significance of differentially expressed genes. **d** Breakdown of regulated sites by sense/anti-sense strands. **e** **Enriched pathways. Pathway significantly enriched have a $-\log_{10}$ (P -value) greater than 1.3 (P -value <0.05 ; Fisher's exact test). Ingenuity canonical pathways analysis in genes with enhanced proximal and distal sites.** **f** Relationship between 3'UTR shortening and lung cancer survival. *, $P<0.05$; **, $P<0.01$; ***, $P<0.001$; ****, $P<0.0001$.

Page 7: Transcripts with longer 3'UTRs were generally enriched in metabolism and p53 signaling-related pathways (Figure 2e) (**Supplementary Table 7**), collectively suggesting that APA contributes to the molecular features of lung cancer.

2) In Figure 2F, can the authors explain whether the survival plot is based on APA of a single gene (if so, which gene) or based on APA of the 12 identified prognostic genes combined?

We are happy to clarify this point and would also like to first correct a typo in this section, where the number of genes used in the prognostic signature is 21 and not 12.

To determine whether APA captures genes with clinical relevance, we first performed a lasso regression (feature selection step) for each gene, which yielded 21 genes whose PSI values are associated independently with survival (provided in revised Supp Table 7). Using the PSI levels of these 21 genes and their corresponding hazard ratio from multivariate regression correcting for patient age, sex, ancestry and tumor stage as their linear weights (described in Methods and Supplementary Table 7), we generated a prognostic index for each sample in a leave-one-out cross validation. Thus, the Prognosis Index (PI) of k -th sample is defined as,

$$PI = \sum_{i=1}^m \beta_i \chi_i$$

where, β is the cox-risk coefficient of gene i , χ is the gene i PSI value. The patients with a PI greater than the median are categorized as low-risk and the rest as high-risk (Figure 2F). We observed that the patients in the high-risk group have a significantly higher lower survival than patients in low-risk (Log Rank $P<0.001$). We have added a description of these steps used in the “**Identification of clinically relevant APA events**” section in Methods.

Revision in text

Page 24: **Identification of clinically relevant APA events**

To determine whether APA captures genes with clinical relevance, we first performed a lasso regression (feature selection step) for each gene, which yielded 21 genes whose PSI values are associated independently with survival. Using the PSI levels of these 21 genes and their

corresponding hazard ratio from multivariate regression correcting for patient age, sex, ancestry and tumor stage as their linear weights, we generated a prognostic index for each sample in a leave-one-out cross validation. Thus, the Prognosis Index (PI) of k -th sample is defined as,

$$PI = \sum_{i=1}^m \beta_i \chi_i$$

where, β is the cox-risk coefficient of gene i , χ is the gene i PSI value. The patients with a PI greater than the median are categorized as low-risk and the rest as high-risk (Figure 2f).

3) Can the authors give some example scatterplots of gene-wise correlation between PSI and expression?

The Spearman's coefficient is negative means PSI increases when gene expression decreases, which seems to contradict the existing knowledge that 3'UTR shortening (larger PSI) is associated with higher abundance of mRNA (higher expression). Could the author better explain the relationships?

The reviewer raises an important point regarding the relationship between the mRNA expression and PSI correlations. Across the 3,531 genes for which we observed a relationship between APA and tumor/normal status, we observed either strong, moderate or no correlation in both a positive and negative direction. The histogram of this correlation coefficient is provided below.

When a miRNA engages with its cognate miRNA binding site, it can induce message degradation or destabilization. Thus, escape from miRNA repression *can* result in a more stable transcript with longer half-life and relative higher abundance of the mRNA, resulting in more protein^{2,3}. Alternatively, mRNA levels are not always affected given that most miRNA/mRNA binding interactions are imperfect and do not lead to mRNA degradation (unlike in plants where the perfect complementarity leads to message degradation). Other studies have shown how loss or mutation of a miRNA binding site, while not affecting mRNA, can affect protein. Lastly, the

ceRNA hypothesis argues that, rather than affecting mRNA expression *in cis*, APA impacts mRNA *in trans* by altering the stochastic nature by which the loss of one miRNA binding site frees up miRNAs to bind to other targets^{2, 7, 8, 9, 10, 11}. Overall, our data reflect the myriad of ways in which 3'UTR shortening can affect mRNA levels and how that there isn't one way by which 3'UTR shortening can impact mRNA and protein. Indeed, as mentioned, one additional factor driving stronger protein and mRNA correlations is the abundance by which a protein is expressed. Our manuscript also excluded reference to ALU regions embedded within 3'UTRs, which also impact the stability and expression of a mRNA and which, equally to miRNAs, can be impacted by 3'UTR shortening^{12, 13}. We have tried to expand on these themes in the revised version of the manuscript to give the reader a fuller, and hopefully clearer, view of our findings.

Revision in text:

Page 8: When a miRNA engages with its cognate miRNA binding site, it can induce message degradation or destabilization. Thus, escape from miRNA repression can result in a more stable transcript with longer half-life and relative higher abundance of the mRNA and resulting in more protein^{2, 3}. Alternatively, mRNA levels are not always affected, given that most miRNA/mRNA binding interactions are imperfect and do not lead to mRNA degradation. Lastly, the ceRNA hypothesis argues that, rather than affecting mRNA expression *in cis*, APA impacts mRNA *in trans* by altering the stochastic nature by which the loss of one miRNA binding site frees up miRNAs to bind to other targets^{2, 7, 8, 9, 10, 11}. To test these hypotheses in lung cancer, we first found that among genes with significant 3'UTR shortening, there was an average loss of nine miRNA binding sites per gene (range 1-51), with enrichment for lung cancer-associated miR-124, miR-181, let-7 and miR-27 binding sites (Supplementary Figure 4a & 4b) (Supplementary Table 10 & 11). We calculated a gene-wise correlation between PSI and expression and found that the PSI of 1,376 of the 3,351 genes significantly correlates with expression (Supplementary Table 12), roughly evenly split between positive and negative associations. We then asked, what is the probability of observing this number of genes (N=1,376) or greater by chance. To test this, we shuffled the APA matrix 10,000 times and calculated an empirical significance ($P < 0.17$) by counting the number of times the genes with Spearman Rho > 0.1 are greater than or equal to 1,376.

Page 15: The effect of APA on gene expression, not yet fully understood, is debated and likely multi-modal. One hypothesis is that expression of a shorter 3'UTR leads to increased mRNA¹⁴ or protein^{2,9,10} *in cis*, a type of gene amplification in the absence of mutations or somatic alterations¹⁵. Our data indicate that genes with short mRNA transcripts have a tighter correlation with their respective protein products. This observation agrees with the decrease in concordance between mRNA and protein abundance in some tissues^{16,17} and is consistent with studies showing a stronger gene-wise mRNA/protein correlation in tumor cells than in normal cells¹⁸. Others argue that the selection of short mRNA 3'UTRs through APA is enriched in transcripts predicted to act as competing-endogenous RNAs (ceRNAs) for tumor-suppressor genes¹¹. **These associations are also complicated by the incomplete complementarity between miRNAs and mRNAs in mammals and the presence of other RNA stability regions in 3'UTRs, such as ALU elements. The latter also impact the stability and expression of a mRNA and which, equally to miRNAs, can be impacted by 3'UTR shortening^{12,13}. Further, as our data show, increased use of a proximal APA site in tumor need not necessarily lie in the 3'UTR and there are many instances of increased proximal APA use occurring in an intron or exon. This kind of APA may not impact RNA or protein expression *per se*, but would almost surely impact the protein type produced. Thus, the ultimate impact of APA on the transcriptome and proteome is likely dependent on a multitude of factors.**

4) Can the authors better explain the reasons why they perform the steps in line 224-line 226?

We are happy to clarify this. We calculated a gene-wise correlation between PSI and expression and found that the PSI of 1,376 of the 3,351 genes significantly correlates with expression (Supplementary Table 12). In regard to the above finding, we then asked what the probability is of observing these many genes (N=1,376) or greater by chance. To test this, we shuffled the APA matrix 10,000 times and calculated an empirical significance ($P < 0.17$) by counting the number of times the genes with Spearman Rho > 0.1 are greater than or equal to 1,376.

5) I am not sure why the authors include supplementary Fig. 5A (BRCA dataset) since the manuscript is about lung cancer.

We are happy to expand here on our rationale for including the breast cancer (and TCGA) datasets in our manuscript. The section reviewer is referring to describes the analysis where we tested whether the increased mRNA stability of short isoforms could drive a higher concordance between mRNA-protein levels for genes with shorter 3'UTRs in patient tumors. To perform this analysis, we needed matched PSI, mRNA and protein abundance information from a reasonable number of samples. Surveying across literature, we can only find this information for the samples in TCGA BRCA cohort, where the proteomics is performed via the NCI project called CPTAC. Thus, we performed this analysis in BRCA. There were limited lung cancer data to do this analysis.

6) Can the authors explain the scatterplot in Figure 3A? what does the dots represent? (if it represents host miRNAs, then isn't there supposed to be 8 points instead of 10?) In addition, what is the correlation coefficient and p-value? 7) Similar concerns with Figure 3B.

In Figure 3a, each point within the scatter plot reflects a miRNA where we plotted delta PSI in tumor and normal samples against delta miRNA expression in the same tumor and normal samples. The graph shows that there is a correlation between the length of a parent miRNA transcript (as determined by APA site usage) and the expression of that miRNA in a tumor. The correlation coefficient reflects this correlation for those 8 miRNAs and the *P*-value reflects the statistical significance of that correlation. The Reviewer is correct that there were initially 10 miRNAs where we saw significant differences in polyA site usage between tumor and normal tissues. On page 12, we explain that of the 10 miRNA host genes that underwent recurrent APA in lung cancer, two (miR-3936, miR-646) did not have mapped reads in the small RNAseq file and therefore we could not conduct the miRNA expression correlation. To make this clearer to the reader, we have also added that point to the legend corresponding to the figure.

Revision in text:

Figure 3: Alternative Polyadenylation of Non-coding RNAs in Lung Cancer and Relationship with Expression. **a** Panels show the correlation between the average 3'UTR length difference in tumor and adjacent non-involved tissues (delta PSI) to the average miRNA expression difference in tumor and adjacent non-involved tissues, respectively. In the scatter plots, the x-axis denotes the delta PSI and the y-axis denotes delta miRNA **b** Correlation between delta PSI to the average of long-non-coding RNA expression. **Of the 10 miRNA host genes that underwent recurrent APA in lung cancer, two (miR-3936, miR-646) did not have mapped reads in the small RNAseq file and therefore we could not conduct the miRNA expression correlation. Of the long non-coding RNAs, we had mapped reads expression for 51.** **c** Difference in HuR/ELAV1 binding to mRNA segments gained and lost through APA based on RNA immunoprecipitation (RIP-seq) assays profile measuring ELAV1/HuR RNA binding of K562 leukemia cells and normal GM12878 cells as a control from GSE35585. For a given gene or element, ELAV1/HuR binding is quantified by counting the number of reads between primary proximal and distal polyA site of usage in both cell lines individually. Here in the box plot, the center lines denote the median, the box indicating the interquartile range and the black line represents the rest of the distribution, except for the points that are determined to be "outliers", 1.5 times the interquartile range.

8. Can the author explain what is the shortening in EA and AAs referring to in TCGA 5' - 3' RNA-seq from non-small cell lung cancer? What are the genes in Tumors being compared to since adjacent non-involved samples are not included in this part of the study?

We are happy to try and clarify this. The Reviewer is correct that TCGA used 5'-3' RNAseq and therefore, is under-powered to assess APA in a comprehensive manner. However, as our dataset is the only 3'-5' RNAseq dataset in human cancer tissues that we were aware of, we were eager

to try and at least validate some of our findings, where possible. In TCGA therefore, we computed a mean PSI level across all the genes (mean-PSI) for each tumor sample. For each gene, PSI is a measure of 3' UTR shortening, defined as the ratio of number of reads from proximal sites compared to total reads. As the Reviewer mentioned, TCGA has an additional limitation where the matched normal samples are not frequently available. In fact, we think that the inclusion of the same number of tumor and non-involved tissues is a strength of our study. Thus, in this case, we alternatively asked whether the median PSI from tumors of AAs is higher than EAs.

Minor concerns:

1. This paper still needs some proofreading. For example, the author put an extra “in” in the title. In addition, in line 110, the author mistakenly added a “not” to their main conclusion...

We apologize for these errors and have now carefully proofread the manuscript to rectify all possible errors, including those pointed out by the reviewer.

2. The figures need to be in clear format, some of the figures are very blur in this version of the manuscript.

To incorporate this feedback, we have provided high quality images of each figure accompanying the manuscript and hope that it is now improved.

Reviewer #3 (Remarks to the Author): Expert in health disparities and genetics

In this manuscript, the authors explore the functional role of alternative polyadenylation in determining transcriptome heterogeneity in lung cancer. I have reviewed the manuscript and have the following comments regarding their population:

1) The authors have entitled their manuscript “A comprehensive map..... in a racially balanced African American and European American lung cancer cohort”. It is unclear what the authors mean by “racially balanced”. The inclusion of both African descent and European descent samples does not make this a “racially-balanced” study as the US population consist of many other race/ethnic groups – this study includes only a small sample from two race/ethnic groups. Therefore, I would advise that the authors remove the term “racially-balanced” from their title and abstract.

We agree with the reviewer that the title of our paper could be misleading to some readers. We have therefore changed it to *A comprehensive map of alternative polyadenylation in African American and European American lung cancer patients.*

2) In addition, the authors should indicated that this is a case study and that the subjects

were derived from a case-control study as the authors have stated in line 572 - 573. The use of the word “cohort” in the title can be misleading.

We agree with the above comment and the title has been revised to accommodate this feedback.

New title: A comprehensive map of alternative polyadenylation in African American and European American lung cancer patients

3) The authors also do not specify whether the samples they used in their study are self-identified as African American or European American or if they (the investigators) have ancestry informative markers (AIMs) on these samples. It would be advisable that the authors make clear that they do have AIMs on their samples in their method section. If not, the authors should indicate that these samples are from subjects who self-identified as belonging to one race/ethnic group. Lack of AIMs will weaken the results that looked at differences in alternative polyadenylation by ancestry.

We agree with the reviewer and are happy to provide more details in this section. Each patient was initially included based on self-reported race. This has now been updated in the methods section. Further, as African Americans are an admixed population, we have also cited further evidence from previous work we have done ^{5,6} using ancestry informative markers, which show that the median African ancestry in our population is 77%.

Revision in text

Page 19: The population accrual and eligibility criteria for this study were previously described ^{7,8}. Briefly, the Institutional Review Boards of the National Cancer Institute and the University of Maryland Medical System approved the study and written informed consent was obtained from each participant. **Individuals in this study self-reported as either African American or European American. Our previous work with this population indicates that the median West African ancestry in self-reported African Americans is 77% ^{5,6}.**

4) The authors should note that race is a social construct and use of the word race within the context of this study is not appropriate. The authors should change “race” to “ancestry” in lines 332, 336 (and any other areas where “race” was used).

We agree with the Reviewer that race reflects a social construct, and in the context of disparities, race, perhaps more so than ancestry, plays an important role. It is the context of race within its refines of a social construct which could impact the differences in biology that we observe here. More and more evidence point to a role for socioeconomic status and the built environment in the determinants of health. While in many cases these determinants are linked with access to care, there is a considerable body of literature that also points to the effects of stress on normal and cancer biology. Our study did not find a role for genetics/ancestry on APA. Further, as race was self-reported by the participants in our study, collectively that is why we presented it in this way.

However, the Reviewer makes an important and relevant point that we should further elaborate on in the manuscript. Indeed, the recommendation is in line with the recent NEJM article that calls on manuscripts to explain the presentation of race, ethnicity or ancestry in each study, and to put results into the context of unknown or unmeasured factors associated with each one¹⁹. To this end, we have extended the limitation section of our manuscript to expand on these points and address the use of race, rather than ancestry, in our study.

Revision in text

Page 18: A limitation of our work regarding population differences in APA that should be considered is our use of self-reported race, rather than genetic ancestry, in the classification of EAs and AAs. Race reflects a social construct, and to the extent that it reflects social determinants of health, it could account for some of the population differences that we observe in our study, albeit in a fashion that we do not yet understand. While we cannot pinpoint social, environmental, or genetic factors that would impact population differences in APA, in line with recent recommendations we have tried explain how our populations were selected¹⁹.

5) The demographic and relevant clinical information on the 96 subjects should be included in the manuscript or supplemental tables. The authors indicated that it was present in “Table 1” in line 465 in “Material and Methods” section but I do not see it anywhere in the manuscript or supplementary tables excel file.

We apologize that this file was not uploaded. We have now added it to the revision and below.

Table 1: Demographic characteristics of the population

	African American	European American	P-value
Age	63.7 (8.4)	65.2 (9.3)	0.16
Sex			
Male	10 (21.7)	17 (32.7)	0.28
Female	36 (78.3)	35 (67.3)	
Smoking Status			
Never	3 (6.5)	3 (5.8)	0.78
Former	15 (32.6)	19 (36.5)	
Current	25 (54.3)	29 (55.8)	
Missing	3 (6.5)	1 (1.9)	
Histology			
Adenocarcinoma	25 (54.3)	21 (37.5)	0.57
Squamous	14 (30.4)	21 (37.5)	
Other	7 (15.2)	10 (19.2)	
Stage			
I	23 (50.0)	35 (67.3)	0.27
II	14 (30.4)	12 (23.1)	
III	7 (15.2)	3 (5.8)	
Missing	2 (4.4)	2 (3.9)	

Reviewer #4 (Remarks to the Author): Expert in health disparities and genetics

Major comments:

Figure 2f: Details of the association between APA events and survival are unclear. How was low risk and high risk defined for the APA events? For the Kaplan Meier curves, how was the cut-point of high vs low risk determined? It is unclear how the APA pattern for the 12 genes was modeled. Was a 12-gene score developed to summarize all genes? What p-value threshold was used to determine the 12 genes to include?

We are happy to clarify this point and would also like to first correct a typo in this section, where the number of genes used in the prognostic signature is 21 and not 12.

To determine whether APA captures genes with clinical relevance, we first performed a lasso regression (feature selection step) for each gene, which yielded 21 genes whose PSI values are associated independently with survival (provided in revised Supp Table 7). Using the PSI levels of these 21 genes and their corresponding hazard ratio from multivariate regression correcting for patient age, sex, ancestry and tumor stage as their linear weights (described in Methods and Supplementary Table 7), we generated a prognostic index for each sample in a leave-one-out cross validation. Thus, the Prognosis Index (PI) of k -th sample is defined as,

$$PI = \sum_{i=1}^m \beta_i \chi_i$$

where, β is the cox-risk coefficient of gene i , χ is the gene i PSI value. The patients with a PI greater than the median are categorized as low-risk and the rest as high-risk (Figure 2F). We observed that the patients in the high-risk group have a significantly higher lower survival than patients in low-risk (Log Rank $P < 0.001$). We have added a description of these steps used in the “**Identification of clinically relevant APA events**” section in Methods.

Revision in text

Page 24: **Identification of clinically relevant APA events**

To determine whether APA captures genes with clinical relevance, we first performed a lasso regression (feature selection step) for each gene, which yielded 21 genes whose PSI values are associated independently with survival. Using the PSI levels of these 21 genes and their corresponding hazard ratio from multivariate regression correcting for patient age, sex, ancestry and tumor stage as their linear weights, we generated a prognostic index for each sample in a leave-one-out cross validation. Thus, the Prognosis Index (PI) of k -th sample is defined as,

$$PI = \sum_{i=1}^m \beta_i \chi_i$$

where, β is the cox-risk coefficient of gene i , χ is the gene i PSI value. The patients with a PI greater than the median are categorized as low-risk and the rest as high-risk (Figure 2F).

Supp Table 7 appears to have 13 significant genes ($p < 0.05$).

Supplementary Table 7, which is now Supplementary 8 in the revision, outlines the significance and weights of the multivariate regression. Genes identified by the LASSO regression don't necessarily need to have a significant coefficient in multivariate regression as the latter corrects age, sex, race and tumor stage.

Given that the number of genes undergoing shortening is greater in EA than AA, should a different set of genes comprise the 12-gene panel for EA and AA? Does the APA association with survival hold in both race groups?

We chose to not compute the prognostic signature for each ancestry separately as we have a limited number of samples for each (N=49 for each). However, considering this comment, we tested whether the performance of the above prognostic signature is different in EA and AA patients. To this end, we computed the hazard ratio of the prognostic signature for EA and AA patients separately (For AA - Hazard Ratio=4.1, $P<0.001$; For EA - Hazard Ratio=4.3, $P<0.001$) and found no significant difference (Interaction P-value = 0.37). We think these are important details to add to the text and have revised the Results section accordingly.

Revision in text

Page 11: **We also tested the performance of the prognostic index separately in EAs and AAs and computed the hazard ration (HR) for each population separately (AA HR=4.1, $P<0.001$; EA HR=4.3, $P<0.001$).**

2. Clinical features: It appears neither treatment nor race were included in the Cox regression model. Given the known impact of treatment on lung cancer survival, racial differences in lung cancer survival, and the authors demonstration of racial differences in APA events, these are likely important variables to consider. How does the lack of including covariates race and treatment in the Cox model impact the authors findings?

Our study did not have complete information on treatment for our population and therefore we were not able to include it in the model. The majority of patients in this study were stage 1. As most stage 1 patients receive surgery alone (due to a lack of consistent evidence that adding chemotherapy to the treatment regime for these patients provides a survival benefit), it is likely that treatment does not significantly impact these results. The reviewer is correct that self-reported race was not included in the analysis; we therefore repeated the analysis and now include those updated weights in the manuscript (Supplementary Table 7).

Minor comments:

1. Table 1 is referenced in the manuscript but appears to be missing from the files.

Thank you to the Reviewer for catching this. We have now corrected it.

2. Figure 2C is mentioned in the text after Figure 2D.

Thank you to the Reviewer for catching this. We have now corrected it.

3. Whether race was self-reported should be stated in the text. While the authors cite their prior work with descriptive characteristics of the study population, it is appropriate to provide to the reader information details as to how race was ascertained given the manuscript emphasis on racial differences.

We agree with the reviewer and are happy to provide more details in this section. Each patient was initially included based on self-reported race. This has now been updated in the methods section. Further, as African Americans are an admixed population, we have also cited further evidence from previous work we have done ^{5,6} using ancestry informative markers, which both show that the median African ancestry in our population is 77%.

Revision in text

Page 19: The population accrual and eligibility criteria for this study were previously described ^{20,21}. Briefly, the Institutional Review Boards of the National Cancer Institute and the University of Maryland Medical System approved the study and written informed consent was obtained from each participant. **Individuals in this study self-reported as either African American or European American. Our previous work with this population indicates that the median West African ancestry in self-reported African Americans is 77% ^{22,23}.**

Revision in text

Page 18: **A limitation of our work regarding population differences in APA that should be considered is our use of self-reported race, rather than genetic ancestry, in the classification of EAs and AAs. Race reflects a social construct, and to the extent that it reflects social determinants of health, it could account for some of the population differences that we observe in our study, albeit in a fashion that we do not yet understand. While we cannot pinpoint social, environmental, or genetic factors that would impact population differences in APA, in line with recent recommendations we have tried explain how our populations were selected ¹⁹.**

4. Figure 4C. X-axis shows genes yet appears to be mislabeled as Race.

Thank you for outlining this, we have corrected it.

5. Page 4, line 110. Extra word ‘not’ appears and should be removed, i.e. “The greater resolution provided by this 3’ to 5’ sequencing method has not enabled...”

Thank you for outlining this, we have corrected it.

6. Page 10, line 226. Text indicates greater or equal to 1,156. I think the authors are referring to a critical value of 1.156. Change comma to decimal.

Thank you for the opportunity to clarify this. Here, we are not referring to the critical value of 1.156, but reviewer correctly spotted the typo as we were referring to number 1,356. Here is the paragraph which has also been corrected accordingly in the “Results” section.

Revision in text

Page 9: We calculated a gene-wise correlation between PSI and expression and found that the PSI of 1,376 of the 3,351 genes significantly correlates with expression (Supplementary Table 12). **We then asked, what is the probability of observing this number of genes (N=1,376) or greater by chance.** To test this, we shuffled the APA matrix 10,000 times and calculated an empirical significance ($P < 0.17$) by counting the number of times the genes with Spearman $Rho > 0.1$ are greater than or equal to **1,376**.

References

1. Mitchell KA, Zingone A, Toulabi L, Boeckelman J, Ryan BM. Comparative Transcriptome Profiling Reveals Coding and Noncoding RNA Differences in NSCLC from African Americans and European Americans. *Clin Cancer Res* **23**, 7412-7425 (2017).
2. Mayr C, Bartel DP. Widespread shortening of 3'UTRs by alternative cleavage and polyadenylation activates oncogenes in cancer cells. *Cell* **138**, 673-684 (2009).
3. Sandberg R, Neilson JR, Sarma A, Sharp PA, Burge CB. Proliferating cells express mRNAs with shortened 3' untranslated regions and fewer microRNA target sites. *Science* **320**, 1643-1647 (2008).
4. Arauz RF, *et al.* Whole-Exome Profiling of NSCLC Among African Americans. *J Thorac Oncol* **15**, 1880-1892 (2020).
5. Mitchell KA, *et al.* Recurrent PTPRT/JAK2 mutations in lung adenocarcinoma among African Americans. *Nat Commun* **10**, 5735 (2019).
6. Xia Z, *et al.* Dynamic analyses of alternative polyadenylation from RNA-seq reveal a 3'-UTR landscape across seven tumour types. *Nat Commun* **5**, 5274 (2014).
7. Poliseno L, Salmena L, Zhang J, Carver B, Haveman WJ, Pandolfi PP. A coding-independent function of gene and pseudogene mRNAs regulates tumour biology. *Nature* **465**, 1033-1038 (2010).
8. Zheng L, *et al.* StarD13 3'-untranslated region functions as a ceRNA for TP53INP1 in prohibiting migration and invasion of breast cancer cells by regulating miR-125b activity. *Eur J Cell Biol* **97**, 23-31 (2018).

9. Orkin SH, Cheng TC, Antonarakis SE, Kazazian HH, Jr. Thalassemia due to a mutation in the cleavage-polyadenylation signal of the human beta-globin gene. *EMBO J* **4**, 453-456 (1985).
10. Steri M, *et al.* Overexpression of the Cytokine BAFF and Autoimmunity Risk. *N Engl J Med* **376**, 1615-1626 (2017).
11. Park HJ, *et al.* 3' UTR shortening represses tumor-suppressor genes in trans by disrupting ceRNA crosstalk. *Nature genetics* **50**, 783-789 (2018).
12. Fitzpatrick T, Huang S. 3'-UTR-located inverted Alu repeats facilitate mRNA translational repression and stress granule accumulation. *Nucleus* **3**, 359-369 (2012).
13. Farre D, Engel P, Angulo A. Novel Role of 3'UTR-Embedded Alu Elements as Facilitators of Processed Pseudogene Genesis and Host Gene Capture by Viral Genomes. *PLoS One* **11**, e0169196 (2016).
14. Zheng D, *et al.* Cellular stress alters 3'UTR landscape through alternative polyadenylation and isoform-specific degradation. *Nat Commun* **9**, 2268 (2018).
15. Venkat S, *et al.* Alternative polyadenylation drives oncogenic gene expression in pancreatic ductal adenocarcinoma. *Genome Res*, (2020).
16. Kosti I, Jain N, Aran D, Butte AJ, Sirota M. Cross-tissue Analysis of Gene and Protein Expression in Normal and Cancer Tissues. *Sci Rep* **6**, 24799 (2016).
17. Vogel C, Marcotte EM. Insights into the regulation of protein abundance from proteomic and transcriptomic analyses. *Nat Rev Genet* **13**, 227-232 (2012).
18. Tang W, *et al.* Integrated proteotranscriptomics of breast cancer reveals globally increased protein-mRNA concordance associated with subtypes and survival. *Genome Med* **10**, 94 (2018).
19. Borrell LN, *et al.* Race and Genetic Ancestry in Medicine - A Time for Reckoning with Racism. *N Engl J Med* **384**, 474-480 (2021).
20. Robles AI, *et al.* A DRD1 polymorphism predisposes to lung cancer among those exposed to secondhand smoke during childhood. *Cancer Prev Res (Phila)* **7**, 1210-1218 (2014).
21. Enewold L, *et al.* Serum concentrations of cytokines and lung cancer survival in African Americans and Caucasians. *Cancer Epidemiol Biomarkers Prev* **18**, 215-222 (2009).
22. Mitchell KA, *et al.* Relationship between West African ancestry with lung cancer risk and survival in African Americans. *Cancer Causes Control* **30**, 1259-1268 (2019).
23. Sinha S, *et al.* Higher prevalence of homologous recombination deficiency in tumors from African Americans versus European Americans. *Nature Cancer* **1**, 112-121 (2020).

REVIEWERS' COMMENTS

Reviewer #1 (Remarks to the Author):

The authors have addressed all my concerns.

Reviewer #2 (Remarks to the Author):

The authors have addressed my concerns.

Reviewer #3 (Remarks to the Author):

I would like to thank the authors for addressing the critiques presented to them. However the following concerns remain:

1) The authors still referred to their population as racially balanced in their abstract. This needs to be corrected as indicated in my last critique of the title.

2) The authors have provided a response for their use of a social construct, race, in research that seeks to understand downstream biological events that may have arisen from issues related to SDOH and structural racism. In addressing this limitation the authors state that "A limitation of our work regarding population differences in 440 APA that should be considered is our use of self-reported race, rather than genetic ancestry, in 441 the classification of EAs and AA". It should be noted that ancestry does not denote a classification into a race group, rather it denotes a geographic region. Therefore, I would advise a slight revision of the statement to "A limitation of our work regarding population differences in 440 APA that should be considered is our use of self-reported race in 441 the classification of EAs and AA".

Reviewer #4 (Remarks to the Author):

The authors have addressed my concerns. I have one minor requested edit or clarification. The authors describe (on page 24) a multivariate regression model, corrected for patient age, sex, ancestry, and tumor stage. Yet the authors indicate they did not use ancestry but rather race in their statistical analyses. Can the authors clarify which variable they used in the model? If genetic ancestry was used in the regression model, then a description of how the ancestry estimates were obtained should be added to the manuscript.

Response to Reviewers

Reviewer #3 (Remarks to the Author):

I would like to thank the authors for addressing the critiques presented to them. However, the following concerns remain:

1) The authors still referred to their population as racially balanced in their abstract. This needs to be corrected as indicated in my last critique of the title.

We apologize with the Reviewer for failing to apply the correction previously suggested also in the abstract. The term “racially balanced” has now been removed from the abstract to accommodate this feedback.

Revision in text

Abstract:

Deciphering the post-transcriptional mechanisms (PTM) regulating gene expression is critical to understand the dynamics underlying transcriptomic regulation in cancer. Alternative polyadenylation (APA)—regulation of mRNA 3'UTR length by alternating poly(A) site usage—is a key PTM mechanism whose comprehensive analysis in cancer remains an important open challenge. Here we use a novel method and analysis pipeline that sequences 3'end-enriched RNA directly to overcome the saturation limitation of traditional 5'-3' based sequencing. Using this method, we comprehensively map the APA landscape in lung cancer in a cohort of 98 tumor/non-involved tissues derived from European American and African American patients.

2) The authors have provided a response for their use of a social construct, race, in research that seeks to understand downstream biological events that may have arisen from issues related to SDOH and structural racism. In addressing this limitation the authors state that "A limitation of our work regarding population differences in 440 APA that should be considered is our use of self-reported race, rather than genetic ancestry, in 441 the classification of EAs and AA". It should be noted that ancestry does not denote a classification into a race group, rather it denotes a geographic region. Therefore, I would advise a slight revision of the statement to "A limitation of our work regarding population differences in 440 APA that should be considered is our use of self-reported race in 441 the classification of EAs and AA".

We agree with the Reviewer and we are happy to follow his advice. We have revised our statement as suggested.

Revision in text

Page 19: Also, our analysis filtered out polyA sites without a well characterized hexamer in the upstream pipeline making it possible that we overlooked a small portion of regulated APA

events. A limitation of our work regarding population differences in APA that should be considered is our use of self-reported race in the classification of EAs and AAs.

Reviewer #4 (Remarks to the Author):

The authors have addressed my concerns. I have one minor requested edit or clarification. The authors describe (on page 24) a multivariate regression model, corrected for patient age, sex, ancestry, and tumor stage. Yet the authors indicate they did not use ancestry but rather race in their statistical analyses. Can the authors clarify which variable they used in the model? If genetic ancestry was used in the regression model, then a description of how the ancestry estimates were obtained should be added to the manuscript.

We apologize for this error in terminology. We have used the self-reported race information throughout the manuscript. Thank you to the Reviewer for catching this. We have now corrected it.

Revision in text

Page 25: To determine whether APA captures genes with clinical relevance, we first performed a lasso regression (feature selection step) for each gene, which yielded 21 genes whose PSI values are associated independently with survival. Using the PSI levels of these 21 genes and their corresponding hazard ratio from multivariate regression correcting for patient age, sex, race, and tumor stage as their linear weights, we generated a prognostic index for each sample in a leave-one-out cross validation.